# Coral Reef Mapping of UAV: A Comparison of Sun Glint Correction Methods

**Aidy M Muslim**, **Wei Sheng Chong**, **Che Din Mohd Safuan, Idham Khalil** and **Mohammad Shawkat Hossain \***

Institute of Oceanography and Environment (INOS), Universiti Malaysia Terengganu (UMT), 21030 Kuala Nerus, Terengganu, Malaysia; aidy@umt.edu.my (A.M.M.); gsk2640@pps.umt.edu.my (W.S.C.); p3057@pps.umt.edu.my (C.D.M.S.); idham@umt.edu.my (I.K.)

\* Correspondence: shawkat@umt.edu.my; Tel.: +609-6683195

**Abstract:** Although methods were proposed for eliminating sun glint effects from airborne and satellite images over coral reef environments, a method was not proposed previously for unmanned aerial vehicle (UAV) image data. De-glinting in UAV image analysis may improve coral distribution mapping accuracy result compared with an uncorrected image classification technique. The objective of this research was to determine accuracy of coral reef habitat classification maps based on glint correction methods proposed by Lyzenga et al., Joyce, Hedley et al., and Goodman et al. The UAV imagery collected from the coral-dominated Pulau Bidong (Peninsular Malaysia) on 20 April 2016 was analyzed in this study. Images were pre-processed with the following two strategies: Strategy-1 was the glint removal technique applied to the whole image, while Strategy-2 used only the regions impacted by glint instead of the whole image. Accuracy measures for the glint corrected images showed that the method proposed by Lyzenga et al. following Strategy-2 could eliminate glints over the branching coral—*Acropora* (BC), tabulate coral—*Acropora + Montipora* (TC), patch coral (PC), coral rubble (R), and sand (S) with greater accuracy than the other four methods using Strategy-1. Tested in two different coral environments (Site-1: Pantai Pasir Cina and Site-2: Pantai Vietnam), the glint-removed UAV imagery produced reliable maps of coral habitat distribution with finer details. The proposed strategies can potentially be used to remove glint from UAV imagery and may improve usability of glint-affected imagery, for analyzing spatiotemporal changes of coral habitats from multi-temporal UAV imagery.

**Keywords:** UAV; glint correction; coral; benthic habitat; South China Sea; Malaysia

## 1. Introduction

Coral reef platforms support diverse marine organisms including soft and hard corals, invertebrates, and sponges. They provide many ecosystem services including revenue from fish and fisheries, tourism, and reproductive sites for turtles and birds in shallow coastal environments. Like other coastal and marine environments, corals are currently vulnerable to environmental and anthropogenic threats across local to global scales: overexploitation of coastal resources, diving, thermal stress, reduction in water quality, climate change, and ocean acidification [1,2]. Mapping coral reef environments is an essential activity to help understand their health, monitor changes related to environmental drivers or anthropogenic stress, and recover from stressors [3–5]. In order to efficiently manage and protect coral ecosystems, managers require detailed information on the distribution, extent and health of reef systems. Large-scale (tens of square kilometers) and fine-scale (detailed mapping using pixels <10 m) studies on corals including reefs and their ecological communities require satellite or airborne images with a high spatial resolution. The unmanned aerial vehicle

(UAV) is characterized by better spatial, temporal, and radiometric resolution than any airborne or satellite platform [6–9]. With multispectral and hyperspectral sensors mounted on UAV platforms, high-resolution, georeferenced data can be acquired for studying spatial and temporal changes in water quality [10] and coral state and bleaching [11,12]. UAV missions, however, can be seriously hindered by specular water reflection problems such as hotspots, sun flaring, or sun glint [13].

The emergence and development of both sensor technology and image processing algorithms eventually provided the enhanced capability of remote sensing for coral studies [4]. Multispectral airborne and satellite sensors provide useful information for mapping submerged aquatic vegetation such as coral benthic habitats, algae and seagrass meadows, and their ecosystem services [3,4,14–17] in the shallow coastal areas where the spatiotemporal heterogeneity is high [18,19]. However, aquatic remote sensing can be seriously impeded, mainly by the water quality (depth and clarity), and sun glint effects. The sun glint problem is produced especially in a wind-roughened surface of water [20,21]. When the sun incidence angle is equal to the reflection angle, the incident light reflected on the rough water causes specular reflection onto the camera. Commonly, the specular reflection causes high brightness in images, reduces the signal-to-noise ratio, and results in a significant reduction in valid observations. Thus, the sun glint problem limits the usefulness and accuracy of remotely sensed data [21,22]. Extraction of detailed benthic habitat extent and distribution information from sun glint-affected data is impossible for both single and multi-mission UAV flight acquisition data due to repeated invalid observations in the study area [22,23].

To avoid sun glint, some suggested conducting aerial surveys during an appropriate time of the day, considering the sun (zenith) angle, wind speed, and field of view of the sensor [24]. This may restrict the flying time available. While flying time is preferred during the lowest low tides, especially for benthic habitat surveys, to avoid confounding effects of water columns, the sun glint effect remains a potential problem. In this context, sun glint correction when mapping coral habitats was demonstrated an essential step in the image processing workflow of UAV imagery.

The negative effects of sun glint in ocean color remote sensing and their possible solutions are well documented [22]. The techniques of sun glint correction in ocean color and coastal and inland remote sensing are many (Table S1, Supplementary Materials). Two sun glint removal techniques are commonly used: (1) a radiative transfer model coupled with a statistical model of surface water to predict water leaving reflectance [25,26], and (2) using near-infrared (NIR) wavelengths (700–1000 nm), which exhibit maximum absorption and minimal water leaving radiance over clear waters, as a proxy for the amount of sun glint in a pixel, and finding the spatial variation of glint intensity across the image [27–31]. The developed sun glint rescue approaches are based on statistical analyses (probability distribution) of water surfaces in satellite imagery and are not valid for high-resolution imagery captured from UAV platforms [10].

Although previous studies attempted to improve the image processing workflow in order to correct sun glint defects on images, most of them applied it to satellite and airborne images. The proposed correction methods were applied to (1) ocean color remote sensing imagery with coarse spatial resolution (>100 m each pixel) [25,32–34] or (2) coastal images with high spatial resolution (<10 m each pixel) [28,35,36]. Recently, a few studies addressed sun glint issues in UAV applications, for example, identifying the sun glint situation [13,37], characterizing environmental variables including sun glint on water reflectance [10], and removing sun glint effects for successful survey of marine fauna like dugong [23]. The internal (focal length, principal point, lens distortion) and external orientations (*X, Y, Z* spatial positions and omega, phi, kappa angular positions) indicating stability of the drone are all taken into account before UAV imagery is stitched together by a robust photogrammetric approach. There are few studies of sun glint correction methods applied to UAV camera imagery across coral reef areas. Thus, further studies on the development of sun glint correction methods, particularly for UAV applications over coral waters, are desirable.

As mentioned earlier, when sun glint correction is performed, it is assumed that the water-leaving signal of the pixels with near-zero in the NIR spectrum does not contain glint, and the largest NIR

values (in the deep-water area) mainly contain glint. In heterogeneous coastal environments [38], however, in the NIR spectra, there is often "residual" radiance. The NIR signal could be strong due to the optical shallow-water quality and the presence of underwater substrates, for example, sandy substrate [39], seagrasses [18], and corals [40,41]. Therefore, for coral-reef areas where the water surface is less than 2 m [42,43], the common sun glint correction assumption may not work and necessitates further extensive study [30]. In some cases, for example, the presence of benthic substrates made the NIR signal strong [42,43] and the glint removal technique proved to be inefficient [30]. Apart from poor assumptions, some techniques require ancillary information such as wind speed and direction, wave slope, and other parameters at the time of satellite overpass, which is difficult to manage [20,36]. Therefore, many authors proposed many alternative solutions which do not require ancillary data to precisely predict glint radiance (Table S1, Supplementary Materials). However, each sun glint removal method has its respective merits and shortcomings. It is, therefore, essential to investigate the efficiency of sun glint correction methods that can improve image quality and, thus, enhance the utility of these methods for case-specific scientific applications.

Applications involving the use of UAV camera image pre-processing techniques for sun glint effects in coral reef mapping were not extensively evaluated and implemented. An appropriate sun glint removal technique is yet to be examined to improve UAV imagery, as well as the usefulness of this method for coral and related habitat classification and distribution mapping.

The overall goal of this study was to test and validate a sun glint correction method for UAV imagery that can be used for coral reef mapping applications. The relative efficiency of the optimized algorithm was tested with a view to use high-resolution UAV images to evaluate its potential for improving the conservation and management of coral resources.

## 2. Materials and Methods

The UAV platform used was a fully equipped DJI Matrice 100 quadrotor, a developer-grade drone with a 3-kg payload weight, which endured for about 14 min. A total of 2345 photos were captured covering an area of 41 ha (Site-1: 14 ha and Site-2: 27 ha) of the west coast of Pulau Bidong, off the east coast of Terengganu, Peninsular Malaysia (Figure 1). Pulau Bidong is an uninhabited island with 242 ha of land, and the shoreline of 9 km is surrounded by the South China Sea. Two sites were chosen for this study considering the existence of coral community complexes. Two flight missions were accomplished: one over Site-1, locally known as Pantai Pasir Cina, and the other over Site-2, locally known as Pantai Vietnam, covering 14 and 27 ha, respectively (Figure 1). Although the geographic distance between the two sites was close (about 500 m apart from each other), there were observable differences in coral community assemblages (Table 1).

Note that data acquired from Site-1 were used to evaluate sun glint models, while data acquired from Site-2 (twice as large as Site-1) were used to further validate the Site-1 method.

### 2.1. UAV Data: Acquisition and Processing

The approach to multispectral UAV image acquisition and mapping of coral benthic habitat is outlined in Figure 2. It was based on four main steps: (1) all necessary activities for the multispectral UAV image processing and analysis (data collection missions, image radiometric correction, georeferencing, reflectance orthomosaics, and land masks); (2) four different sun glint corrections were applied based on two different strategies (Strategy-1 and Strategy-2; see Figure 2); (3) image training for coral benthic habitat classification; and (4) finally, in the validation, the efficacy of the sun glint correction method was evaluated in terms of accuracy of the map products. These processes are later described in detail.

The UAV data acquisition was performed on 20 April 2016 between 2:00 and 3:00 p.m. Coordinated Universal Time (UTC) +8 when the sun azimuth was 299.13°. The flight mission was intentionally scheduled at midday so that there was an obvious sun glint effect, which was the primary objective of this study. The mission planner software Litchi (v4.0.1; https://flylitchi.com/) was used to design flight tracks for each site and to monitor progress during data acquisition. Flight missions were flown at

an altitude of 160 m above ground level (AGL) and a cruising speed of 38 km/h. UAV flight tracks were designed in a lawn-mowing pattern to enable expected overlap, with a 75% frontlap and 75% sidelap. The UAV flight starting waypoint was set near the coast, and the subsequent waypoints were away from the coast to the border of fringing reefs. The mission planner allowed the end waypoint to be the landing site, i.e., the first starting waypoint. This design allowed easy retrieval of drones when the weather was unexpected, or following marine bird interference or system failure. Surveys were carried out at low tide (−0.1 m from mean sea level) to ensure that the effect of sun glint exposure on the object of interest was investigated exclusively. The sites exhibit a mixed tide with a dominant diurnal tide [44].

The environmental conditions during the UAV missions and in situ benthic surveys were also recorded (Table 2). The oceanographic data were retrieved from the Hybrid Coordinate Ocean Model, or HYCOM (http://www.hycom.org/), which is a part of the United States (US) Global Ocean Data Assimilation Experiment (GODAE) at a scene position (at 5.60°–5.65° north (N) and 103.0°–103.1° east (E)). HYCOM provides data to produce daily three-dimensional (3D) snapshots of oceanographic variables such as temperature, salinity, and current velocity at 1/12° resolution. On the day of our survey (20 April 2016), the average seawater salinity was 33.1 ppt. Current measurements were taken at two separate flow speeds measured along two orthogonal axes (known as U and V components). The axes were oriented in such a way that U represents the horizontal flow in the east–west direction, while V represents the vertical flow in the north–south direction. The U component was −0.001 m/s, meaning that current flows in the westward direction, and the V component was 0.015 m/s, meaning that current flows in the northward direction.

Meteorological data were recorded from the European Center for Medium-Range Weather Forecasts (ECMWF) online dataset (https://www.ecmwf.int/). ECMWF is an independent intergovernmental organization based in Reading, United Kingdom (UK). During the field trip, the sea surface temperature was about 30.3 °C, and the sky was noticeably clear and sunny (total cloud cover 1.2% at 12:00 p.m. UTC+8) at the beginning of the UAV survey. The cloud cover gradually increased during the benthic in situ data collection time, especially in the evening (total cloud cover 41.9% at 6:00 p.m. UTC+8). The forecasted wind direction and speed suggested that the winds blew westward and subsided near dusk (the U component of wind decreased from noon −3.192 m/s to dusk −2.263 m/s). Furthermore, the wind V component also showed that the wind speed decreased from noon −1.982 m/s to dusk −0.379 m/s (a negative value indicates that the winds blew toward the southern direction of the site).

The tidal predictive data used in this study were taken from the WorldTides web interface (https://www.worldtides.info/) established by US Oregon State University. The WorldTides prediction is based on harmonic analysis, method of least squares (HAMELS). The tide height was taken at 5.621° N and 103.056° E, representing the center position of the scene. During the study periods, tide heights were between −0.1 m and 0.1 (measured from mean sea level). The highest predicted astronomical tide in the island is 1.17 m, and the lowest astronomical tide is −1.13 m. The average of all high waters is 0.47 m, whereas the average of all low waters is −0.47 m from mean sea level. Note that both predicted astronomical tide and mean high/low waters were calculated over a full nodal cycle (~18.6 years), starting at 1 January 2000 by the WorldTides research team.

The UAV was mounted with two cameras: Zenmuse X3 and Micasense RedEdge. The Zenmuse X3 with a rolling shutter and 3.6-mm focal length (https://www.dji.com/) provided 12.0 megapixels (4000 × 3000 pixels) of true-color imagery with non-reflectance-compensated outputs that were was used to get first-person views of the drone's flight mission movements. This study explicitly used the Micasense RedEdge with a global shutter, 47.2° field of view, and 5.5-mm focal length (https://www.micasense.com/) with an image size of 1.3 megapixels (1280 × 960 pixels) for mapping coral habitat. The spectral range of the instrument is 455 to 727 nm with a width of 10–40 nm. The instrument receives radiance in the visible (blue with 475-nm central wavelength, green 560-nm wavelength, and red 668-nm wavelength), near-infrared (NIR; 840-nm wavelength), and red-edge (717-nm wavelength) regions. This camera was pointed near nadir during the missions and programmed in 3 s to capture each

image. Images of a calibrated reflectance panel were captured immediately before and after each flight, along with sun irradiance measurements via an onboard downwelling light sensor to generate reflectance-compensated outputs from Micasense RedEdge. The sky was well lit with low complete cloud cover during the UAV data acquisition (see Table 2).

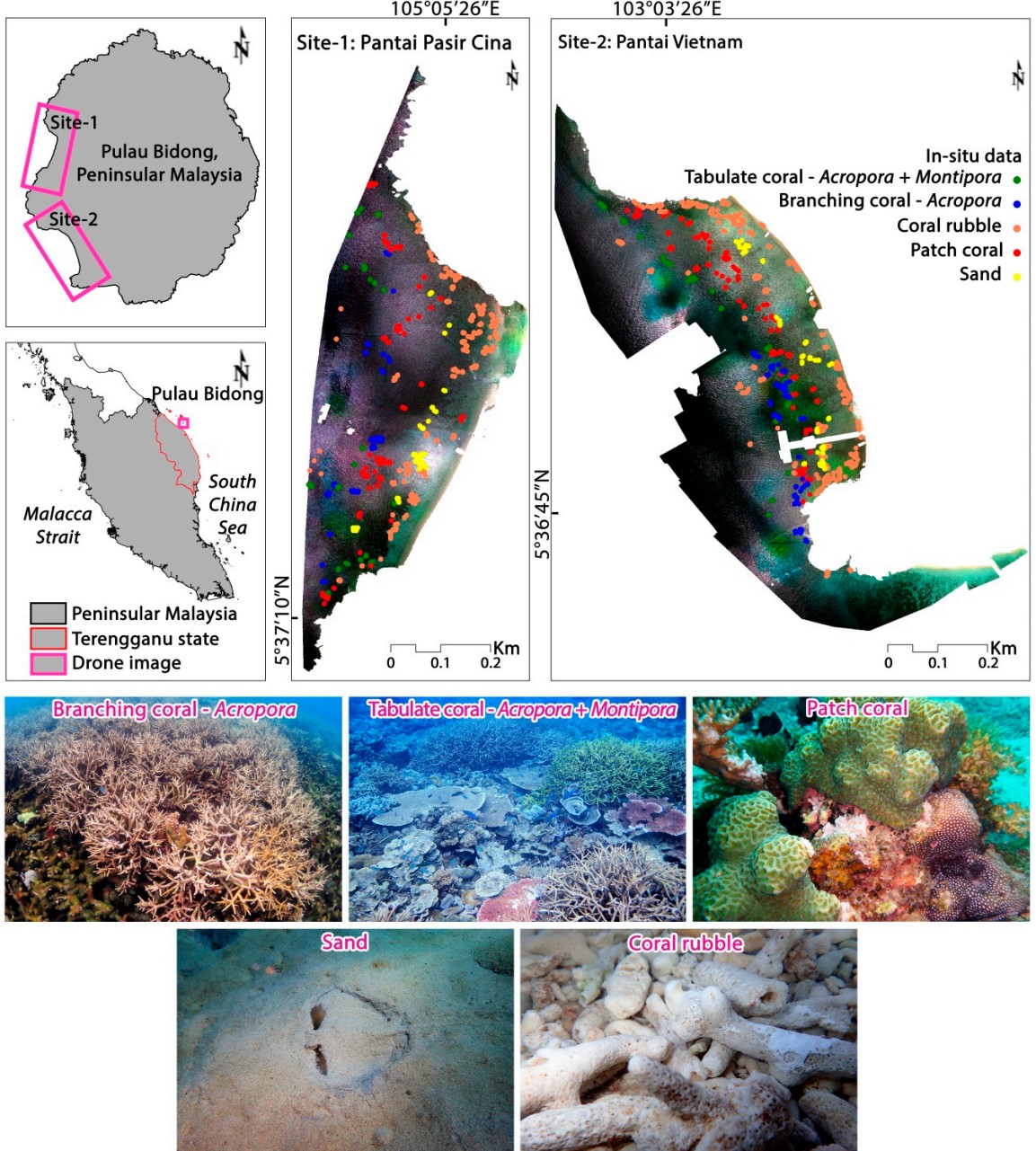

**Figure 1.** Study site map showing Pulau Bidong in Peninsular Malaysia and the locations of in situ data (**top**). Examples of digital photos of each coral habitat class, taken during field surveys, are given below.

**Table 1.** Coral benthic habitat characteristics of study sites.

| Site Characteristics | Site-1: Pantai Pasir Cina | Site-2: Pantai Vietnam |
|---|---|---|
| Tide type | Mixed tide (dominant diurnal) | |
| Coral benthic substrates | | |
| Shallow area (−0.1 to −1 m water depths during the lowest tide) | Fine sand; submerged rocks; coral rubble | Fine sand; submerged rocks; coral rubble |
| Reef flat (−1 to −5 m water depths during the lowest tide) | Live coral cover is dominated by branching coral—*Acropora* (branching and bottlebrush), tabulate coral—*Acropora* and *Montipora*, patch coral, and several non-coral components such as dead coral with algae, coral rubble, sand, and dead mushroom coral | Live coral consists of branching—*Acropora*, tabulate coral—*Acropora*, and massive coral Non-coral cover, such as coral rubble, dead coral with algae, and submerged rocks |
| Reef crest (−5 to −6 m water depths) | Live coral cover is dominated by mushroom coral, branching coral—*Acropora*, and massive coral | Live coral consists of massive coral and tabulate—*Acropora*; submerged rocks are also present |
| Reef front/fore reef (−6 m to −7 m water depths) | Live coral cover is dominated by branching coral—*Acropora* and *Pocillopora* (bushy); another coral colony form is massive coral—*Porites, Favites, Platygyra,* and *Goniastrea* | |
| Deep sea (>−7 m water depth) | Fine sand, dead coral, and live coral patches with a combination of branching, massive, and submassive coral such as *Acropora, Pocillopora, Porites, Favites,* and *Gonisastrea* | Fine sand, dead coral, and live coral patches with the combination of branching, massive, and submassive coral such as *Acropora, Pocillopora, Porites, Favites,* and *Gonisastrea* |

**Table 2.** Oceanographic, meteorological, and tidal data during unmanned aerial vehicle (UAV) missions and in situ benthic surveys. UTC—Coordinated Universal Time.

| Environmental charactEristics | Date: 20 April 2016 | | Data Source |
|---|---|---|---|
| | Time (UTC+8) | | |
| | 12:00 p.m. | 6:00 p.m. | |
| **Oceanographic data** | | | |
| Salinity (ppt) | 33.1 | | Hybrid Coordinate Ocean Model (HYCOM), http://www.hycom.org/ |
| U component of current (m/s) | −0.001 | | |
| V component of current (m/s) | 0.015 | | |
| **Meteorological data** | | | |
| Sea surface temperature (°C) | 30.3 | | European Center for Medium-Range Weather Forecasts (ECMWF), https://www.ecmwf.int/ |
| U component of wind (m/s) | −3.197 | −2.263 | |
| V component of wind (m/s) | −1.982 | −0.379 | |
| Total cloud cover (%) | 1.2 | 41.9 | |
| **Tidal dataat 0.0 m mean sea level** | | | |
| Tide height (m) | −0.1 | 0.1 | |
| Lowest astronomical tide (m) | −1.13 | | WorldTides, https://www.worldtides.info/ |
| Highest astronomical tide (m) | 1.17 | | |
| Mean high water (m) | 0.47 | | |
| Mean low water (m) | −0.47 | | |

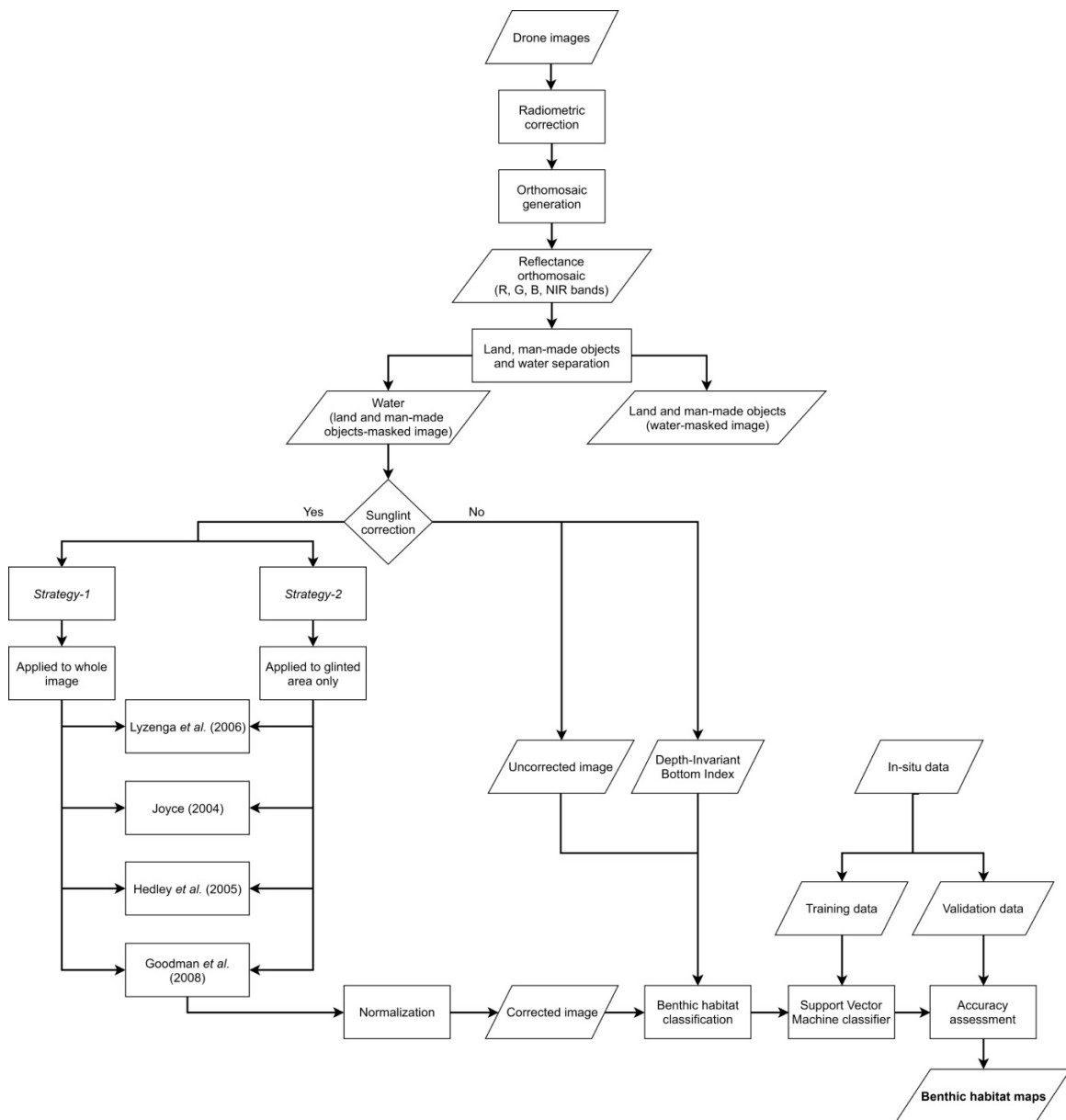

**Figure 2.** Workflow chart for sun glint removal from unmanned aerial vehicle (UAV) data, subsequent coral habitat classification, and product validation.

As suggested in the technical notes of the sensor supplier, the calibration panel was used for radiometric calibration on sunny days (see https://support.micasense.com/hc/en-us/articles/360025336894). The calibration panel has a visible and near-infrared spectrum calibration curve. The sensor supplier provides the calibration data in the range of 400 nm to 850 nm as absolute reflectance (a value between 0 and 1). In this study, the average panel albedo for bands (red = 0.68, green = 0.69, blue = 0.68, NIR = 0.63, red edge = 0.68) was used to represent the calibration curve with five reflectance values or albedos, one for each of the five bands of the Micasense RedEdge camera. The panel dimension was 15.5 cm by 15.5 cm, and, for radiometric calibration, it was ensured that at least one-third of the image width was taken. Due to it being most effective in overcast, completely cloudy conditions, the downwelling light sensor data were not used for image post-processing. The drone global positioning system (GPS) module was installed at a higher position than the light sensor, thus causing its shadow to be cast on the light sensor when the drone changed its cruising direction.

Multispectral images collected from UAV flights were geotagged by an onboard GPS module prior to photogrammetric processing using Pix4D Mapper Pro software (v4.1.24; https://www.pix4d.com/). The radiometric calibration was done automatically in the "DSM (Digital Surface Model), Orthomosaic, and Index" module of Pix4D Mapper Pro and the radiometric correction type selected was "camera only". The corresponding calibration panel image was imported into the module for the selected band, an ROI (Region of Interest) was drawn on the image to define the area of radiometric calibration, and the albedo value was inserted for the selected band. This procedure was repeated for each of the five bands. The module utilized the values of some parameters in the EXIF (Exchangeable Image File Format) metadata of the drone images to correct variables such as incoming sunlight irradiance, ISO, aperture, shutter speed, vignetting, sensor response, and optical system when producing the reflectance map. Further details on the radiometric calibration for the Micasense RedEdge camera can be found at https://support.micasense.com/hc/en-us/articles/115000831714-How-to-Process-RedEdge-Data-in-Pix4D. Each geotagged image was radiometrically calibrated and stitched together to produce a geo-referenced orthomosaic (datum WGS84 UTM zone 48N coordinate system) (Figure 2). The programmable processing settings for the Pix4D orthomosaic generation were as follows: full tie-point image scale, a minimum of three tie-points per image, and a 0.5 image scale with a multi-scale view for point cloud densification. Reflectance orthomosaics (Site-1: 11.25 cm/pixel and Site-2: 11.39 cm/pixel) were imported into ENVI software (https://www.harrisgeospatial.com/) for further spectral analysis in four-band configurations (red (R), green (G), blue (B), NIR). The red-edge band was not used in this study due to it being less useful in coral habitat mapping.

When broadly separate land and water in the masking process, an automatic image processing algorithm could not work due to pixel noise. Therefore, land and water were manually defined by visual interpretation of the false-color composites (band 1 = NIR; band 2 = R; band 3 = G). In false-color composites, water appears blue. Next, the land which included sandy and rocky areas without vegetation was separated from water bodies using normalized difference water index (NDWI) thresholds [45]. The threshold value was positive (between 0 and 1) for water [46]. Human-made objects (i.e., impervious surfaces), such as buoys, jetty, boats, and fish cages, were visually identified using the operator's skill, and their extent was manually delineated. Identified land including vegetation, sand and rocks, and impervious surfaces were masked from the image. The land-masked image underwent sun glint correction to create coral benthic habitat cover maps.

## 2.2. Glint Removal Procedures

Four glint removal procedures were used for image pre-processing (Figure 2), and all are well known for high-resolution imagery of shallow water reef applications [22]. It is worth highlighting that the tested methods are described in terms of reflectance, $R$, but can similarly be applied to data expressed as radiance, $L$, or digital number, $DN$.

Firstly, we applied the three methods proposed by Hedley et al. [28], Lyzenga et al. [35], and Joyce [47]. All these methods used a subset of deep-water pixel samples (see Figure 3 for yellow ROI) to compute the regression slope ($b_i$) from a line of linear correlation between NIR ($R(NIR)$) and each visible band ($R_i(VIS)$) (as expressed in Equation (1)). Note that Hedley et al. [28] and Joyce [47] used the least square approach to define $b_i$, whereas Lyzenga et al. [35] used the covariance between each visible band and the NIR for obtaining $b_i$.

$$R_i(VIS)' = R_i(VIS) - b_i\big[R(NIR) - R_{ref}(NIR)\big], \tag{1}$$

where $R_{ref}$ is the minimum in NIR value used for Hedley et al. [28], the mean NIR value for Lyzenga et al. [35], and the modal NIR value for Joyce [47]. These values were derived from the same set of deep-water pixels.

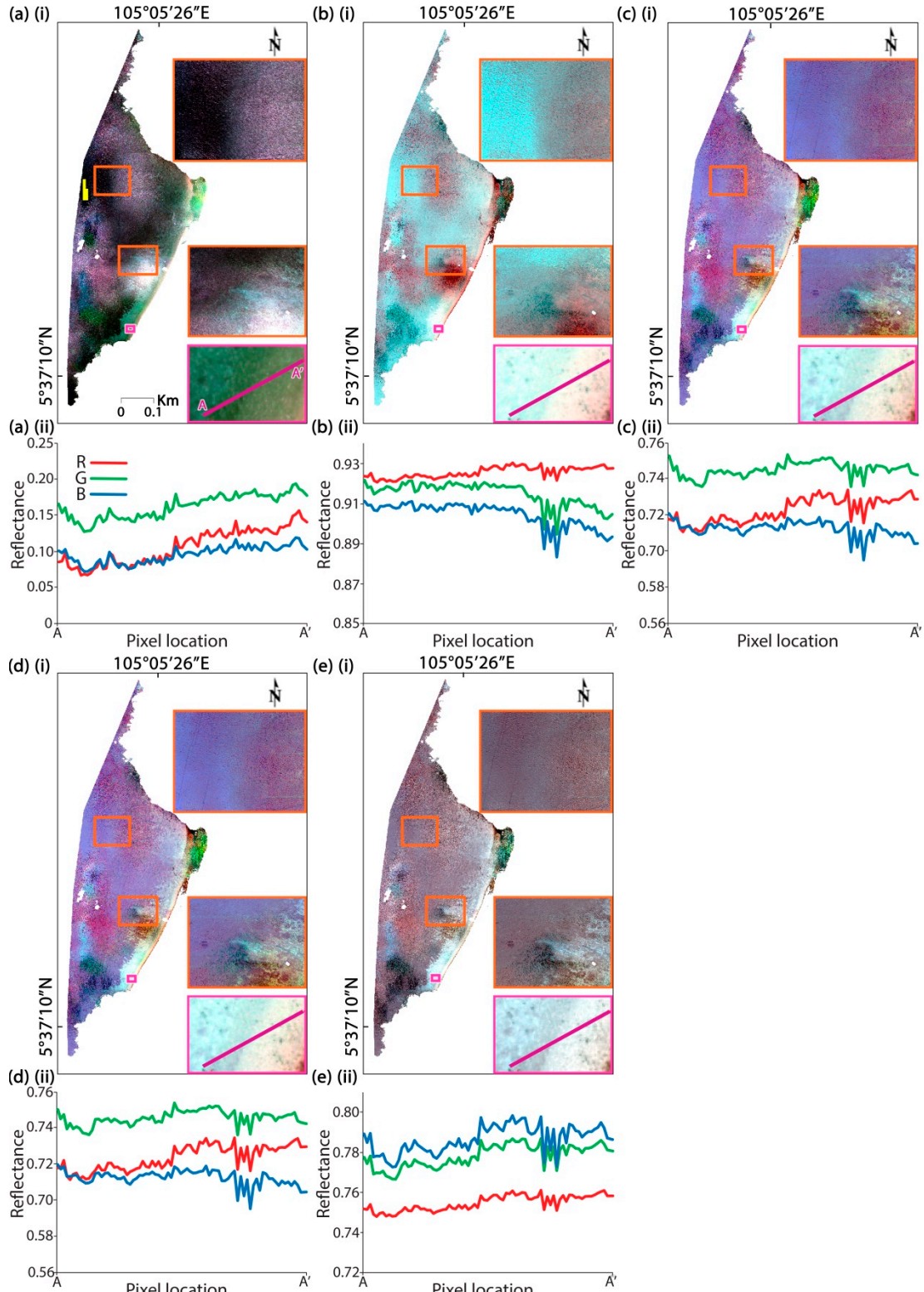

**Figure 3.** UAV images from Site-1 (Pantai Pasir Cina) on 20 April 2016. (**a**) Image without glint correction; images after glint correction using the methods proposed by (**b**) Lyzenga et al. [35], (**c**) Joyce [47], (**d**) Hedley et al. [28], (**e**) Goodman et al. [29], pre-processed using Strategy-1. In (**a**) (**i**) the yellow region of interest (ROI) shows the location of deep-water pixels used for the glint removal procedure (where applicable). In each situation (**i**), a true-color composite with zoomed-in views is shown, along with (**ii**) reflectance spectra of red (R), green (G), and blue (B) bands for pixels along the 10-m transect line from point A (un-glinted area) to A′ (glinted area).

Next, each pixel is corrected independently in the Goodman et al. [29] suggested method. The reflectance in the NIR band is subtracted from each visible band, and a wavelength-independent offset is added. To calculate the offset $\Delta$, the bandwidths tested in the glint removal exercise of Goodman et al. [29] were 640 nm ($R(640)$) and 750 nm ($R(750)$) (Equation (2)). However, the closest bandwidths used in this drone survey, i.e., 668 nm and 840 nm, were employed.

$$\Delta = A + B \left[ R(640) - R(750) \right], \tag{2}$$

where *A* and *B* are constants (*A* = 0.000019 and *B* = 0.1). This study, therefore, applied Equation (3) with slight modification of Goodman et al. [29] in terms of bandwidths.

$$R_i(VIS)' = R_i(VIS) - R(750) + \Delta \tag{3}$$

### 2.3. Evaluation of Sun Glint Correction Methods

The abovementioned four glint removal procedures were applied to images pre-processed with two strategies. Strategy-1 was applied to the whole image. The glint removal procedures were applied to the land-masked image straightway to produce normalized sun glint-corrected images. Zhang and Wang [20] stated that precise knowledge of the performance of different glint removal models across the world's oceans is required to help mask glint-contaminated regions while retaining as much as possible the useful regions. This motivated this study to follow Strategy-2, where glint removal algorithms were applied to the glinted area only. The glinted area in the land-masked image was separated from the un-glinted area via a decision tree (rule-based approach), where several glinted sand pixels were sampled from shallow-water (−0.5 m) and deep-water (−7 m) regions to develop rules to mask the un-glinted areas [13,48]. The glint removal procedures were applied to the glinted area, and the de-glinted area was then merged with the un-glinted area to form normalized sun glint-corrected images. Both strategies were tested on the Site-1 and Site-2 land-masked images to produce sun glint-corrected images (Figure 2). Pixel normalization was performed by adding a positive value to the lowest pixel value among the bands and then dividing by the highest pixel value to set the pixel value range 0 to 1.

To investigate the relative performance of sun glint correction algorithms used for coral benthic habitat mapping, the two strategies with four glint removal methods and their products were compared. The uncorrected image was added to act as a control. Reconstruction of the glint-contaminated area under Strategy-2 and subsequently making the image glint-free may represent a better solution to the glint removal problem. The knowledge rendered from this study may enhance the ability of researchers to handle glint-contaminated remote sensing imagery.

### 2.4. Coral Habitat Mapping

Field data collected during this study were used to train UAV image data (to produce benthic habitat classification maps) and to verify the sun glint-corrected UAV results. Field data were collected from 3:30 to 5:00 p.m. UTC+8 immediately after the UAV acquisition on 20 April 2016. The spot check survey technique [49] was used to collect field data on the benthic composition of coral reefs (Figure 1). Spot check data were recorded either from a boat (depth sounding via a portable depth sounder) or in water (snorkeling and diving) and geotagged (datum WGS84 UTM zone 48N coordinate system) using GPS and Garmin BaseCamp software (available at https://www.garmin.com/). Coral communities were visually identified (Table S2, Supplementary Materials) based on expert knowledge. Each underwater photograph was inspected visually, and literature was used to confirm the benthic class [50]. The in situ data were stratified randomly per site into a training dataset (~60%) and validation dataset (~40%) for use in the classification of benthic habitats. In total, Site-1 in situ data consisted of 147 branching coral—*Acropora* (BC), 157 tabulate coral—*Acropora + Montipora* (TC), 247 patch coral (PC), 329 coral rubble (R), and 91 sand (S); Site-2 comprised 163 BC, 169 TC, 239 PC, 357 R, and 95 S field data points.

Using depth-invariant bottom index (DII) [51,52], a water-column correction technique for both glint-corrected and uncorrected UAV images was performed to remove benthic habitat reflectance variations, as the water depth may affect the classification results (Figure 2). DII requires the ratio of the water-column attenuation coefficient between visible band pairs, which can be derived statistically by using the reflectance of similar objects at different depths. Pixels from water depths of −0.5 m and −3 m covering sand only were sampled to derive three DII combinations from three visible bands (blue and red band; red and green band; green and blue band).

For the benthic habitat image classification, the in situ data were transformed into regions of interest (ROIs) and categorized into five classes by referring to the benthic classification scheme (see Table S2, Supplementary Materials). The training ROIs were overlaid on the sun glint-corrected (following Strategies 1 and 2), uncorrected true-color (RGB) composites and DIIs for each site. Benthic habitat maps were produced using pixel-based image classification via the support vector machine (SVM) algorithm in ENVI software.

SVM is a non-parametric supervised machine learning classifier developed by Cortes and Vapnik [53]. The objective of applying SVM was to find an optimal hyperplane that could separate the input dataset into five benthic classes in a fashion consistent with the training dataset. Due to its high classification accuracy, this method is prescribed for the classification of multispectral images with small separable spectral values and is, thus, suitable for benthic habitat mapping [54]. In SVM, complex hyperplanes are represented by kernels. The Gaussian radial base function kernel is used in SVM for image classification due to its robust capabilities compared to other kernels (e.g., linear, polynomial, and sigmoid kernels) and requires minimal optimization for training [55].

### 2.5. Validation of Sun Glint Correction Methods

To evaluate the performance of using glint removal algorithms on UAV images with an aim to produce coral habitat maps, the quality of the resulted maps was examined through (a) visual inspection and (b) quantitative accuracy assessment [56]. For the accuracy assessment, the benthic habitat maps generated from Strategy-1 and Strategy-2 were compared with independent in situ reference (validation) data not used in training image classification. By applying the "confusion matrix using ROIs" module in ENVI software [57], confusion matrices were developed based on absence or presence using validation ROIs. The matrices provided essential accuracy assessment parameters such as the overall accuracy (OA), user and producer accuracies (UA and PA), and Kappa coefficient of agreement ($k$). These accuracy components take the non-diagonal attributes into interpretation and explain the differences between the actual and expected agreement by chance [58].

## 3. Results

### 3.1. Comparison of Image Quality Based on Glint Correction Methods

To assess how well the different glint correction methods performed in mapping benthic habitats, the results were quantitatively estimated based on the accuracy of the classification of the coral cover, and the products were qualitatively evaluated by visual inspection of RGB composites. Note that the RGB images were radiometrically corrected before the glint removal algorithm was applied to produce reflectance values. All the glint-corrected images were normalized to provide a simulated positive 0 to 1 water-leaving reflectance value. The reflectance values in Figures 3 and 4 are not the "true" benthic reflectance; normalization was performed to observe the effects of glint removal on reflectance. Therefore, the observed differences between those corrected images can be attributed to the sun glint removal techniques only.

The uncorrected image (Figure 3a(i)) exhibited conspicuous sun glint patterns with higher glint intensity in the right side of the image. Winds during image acquisition may explain the higher amount of glint on the right-hand side of the image, as the southern and east parts were sheltered from wave actions. Visual inspection of glint-corrected images suggested that glint cover noticeably hindered the

visual quality of the image and glint removal techniques following Strategy-1 and Strategy-2 improved it (Figures 3 and 4). Glint correction algorithms recovered glint-masked areas, as clearly noticeable in the glint-corrected results (Figures 3b–e and 4b–e).

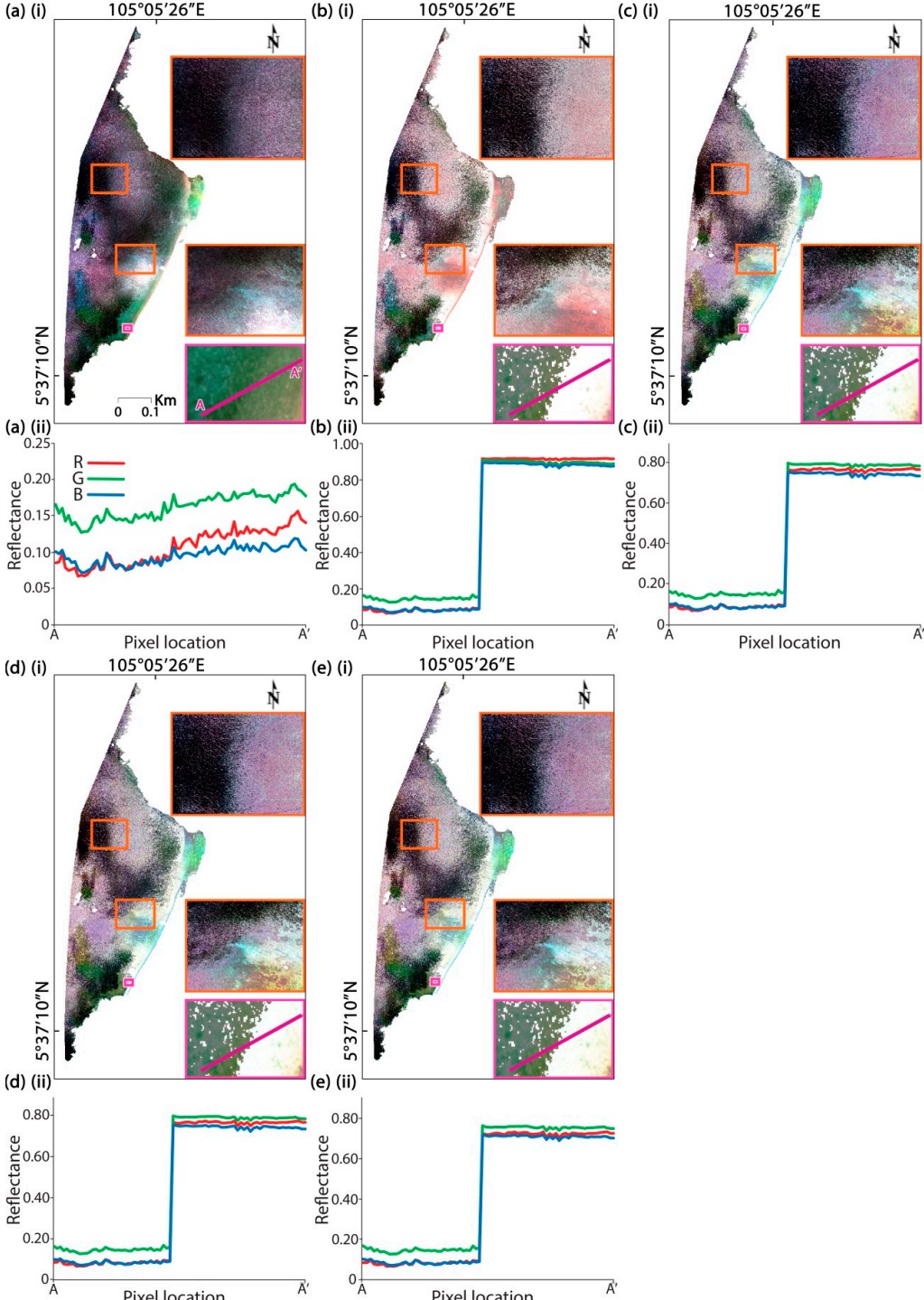

**Figure 4.** UAV images from Site-1 (Pantai Pasir Cina) on 20 April 2016. (**a**) Image without glint correction; images after glint correction using the methods proposed by (**b**) Lyzenga et al. [35], (**c**) Joyce [47], (**d**) Hedley et al. [28], (**e**) Goodman et al. [29], images pre-processed using Strategy-2. In each situation (**i**), a-true color composite with zoomed-in views is shown, along with (**ii**) reflectance spectra of red (R), green (G), and blue (B) bands for pixels along the 10-m transect line from point A (un-glinted area) to A′ (glinted area).

Figure 3b(i)–e(i) show that all four methods following Strategy-1 improved the UAV images from Site-1. The glint removal algorithm removed the discontinuities in glint distribution. The result suggests that the methods proposed by both Hedley et al. [28] and Joyce [47] (Figure 3c(i),d(i)) corrected the glint-impacted pixels in near-shore and very shallow water areas, leaving similar patterns of sun glint, visible as a yellow hue in the near-shore areas and dark purple in deeper areas. The distinctive strong red region found in the image processed using the method proposed by Lyzenga et al. [35] (Figure 3b(i)) indicates that, using Strategy-1, this method could be useful in separating glinted areas from surrounding un-glinted water pixels. The method proposed by Goodman et al. [29] (Figure 3e(i)) also showed consistent sun glint patterns in the shallow-water pixels.

The corrected images also showed variations in the spatial distribution of overcorrection with glint removal methods (Figures 3 and 4). Overcorrection was apparent in the near-shore and shallow areas when all glint removal procedures were applied under Strategy-1, which effectively made the underwater substrates more visible in the sun-spot regions (zoomed-in regions in Figure 3b–e(i)). The method proposed by Goodman et al. [29] (Figure 3e(i)) smoothed out the higher amount of glint areas showing a violet hue in the uncorrected image (Figure 3a(i)). The un-glinted regions in this image, however, were comparable in tone to the glint regions that resulted in transect reflectance spectra being almost flat (Figure 3e(ii)). The presence of white sun spots with less noise coming from underwater substrates also shows that Strategy-2 (zoomed-in regions in Figure 4) was better than Strategy-1.

There was relatively more white noise in the northern parts of the images generated using Strategy-1 compared to the images generated using Strategy-2 (Figure 4). Strategy-2 with the methods proposed by Hedley et al. [28], Joyce [47], and Goodman et al. [29] generated similar results in terms of tone (Figure 4c(i)–e(i)). Strategy-1 can be assumed to treat shallow-water pixels as glinted pixels near Site-1's north-eastern coast, while Strategy-2 (Figure 4b(i)–e(i)) was not impacted by the shallowness of the water.

### 3.2. Spectral Analysis Using UAV Data

Spectral analysis was performed following two approaches: (1) extracting reflectance spectra from small areas in every image with homogeneous bottom type and at a constant water depth, and (2) comparing benthic spectra between uncorrected and corrected images at different depths.

For the first approach, a 10-m-long transect line was drawn over a homogeneous area (covering coral rubble) with a depth of −1 m to examine spectral characteristics, collected from un-glinted to glinted regions, extending from point A to point A′ (see the zoomed-in views in Figures 3 and 4). The reflectance graph for uncorrected image (Figure 3a(ii)) showed an increasing trend from un-glinted point A to glinted point A′, indicating an increase in reflectance spectra mainly due to glint intensity. After applying the glint removal Strategy-1, the reflectance patterns were found to be uniform (Figure 3b–e(ii)) across the transect, and the reflectance spectra collected from glinted areas were hardly distinguishable from un-glinted areas. However, when Strategy-2 was applied (Figure 4), peaks and sharp increases in reflectance spectra were noticeable near the midpoint of the transect. Since the glint removal procedure using Strategy-2 was applied only to the glinted regions, the reflectance in glint contamination-free areas (near point A in Figure 4b–e(ii)) and the uncorrected image (Figure 4a(ii)) gave nearly the same results.

In the sampled shallow water pixels (Figure 3c(ii),d(ii) and Figure 4c(ii),d(ii)), there was a good match between the reflectance spectra obtained using the methods proposed by Hedley et al. [28] and Joyce [47]. As expected from the visual inspection of RGB composite results, the method proposed by Lyzenga et al. [35] using Strategy-2 provided a better image with less noise in near-shore areas, which presented a more bluish tone than others (Figure 3b(ii) and Figure 4b(ii)). The reflectance obtained by the uncorrected image was less than 0.2, while others using Strategy-1 had values greater than 0.58. The high correction factor proposed by Lyzenga et al. [35] may have caused relatively higher reflectance spectra as illustrated in Figure 3b(ii) and Figure 4b(ii).

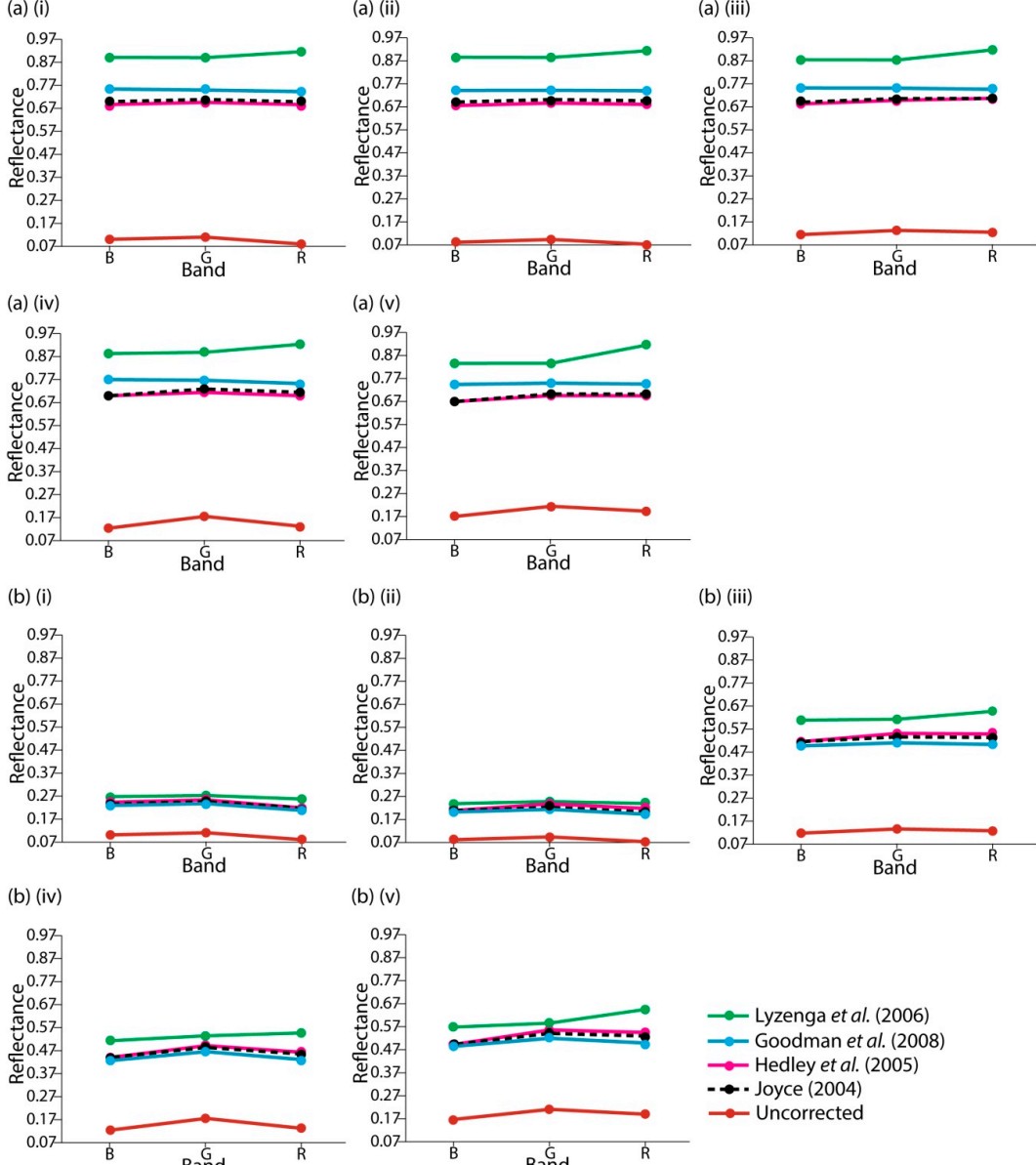

**Figure 5.** Benthic reflectance spectra of Site-1 (Pantai Pasir Cina) UAV image before (uncorrected) and after glint removal using four different methods proposed by Lyzenga et al. [35], Goodman et al. [29], Hedley et al. [28], and Joyce [47], following two strategies: (**a**) Strategy-1 and (**b**) Strategy-2. The benthic spectra were collected from an area with a water depth of approximately −1.5 m, present in the reflectance graphs shown for (**i**) branching coral—*Acropora* (BC), (**ii**) tabulate coral—*Acropora + Montipora* (TC), (**iii**) patch coral (PC), (**iv**) coral rubble (R), and (**v**) sand (S).

For the second spectral analysis approach, field point data (see Table S2, Supplementary Materials) on benthic cover classes were used to collect the reflectance spectra at a water depth of about −1.5 m. The high reflectance values obtained using the method proposed by Lyzenga et al. [35] in the red band (R) (see Figure 5a,b) may explain why some parts of the corrected RGB composites appeared strongly red in Figures 3b and 4b. The uncorrected reflectance spectra were used to investigate the over- or undercorrection due to the glint removal procedure. A near-one reflectance spectrum is an indication of overcorrection. Comparing reflectance spectra collected using Strategy-2 in BC (Figure 5b(i)) and TC (Figure 5b(ii)) shows that those generated with the methods proposed by Lyzenga et al. [35], Hedley et al. [28], Goodman et al. [29], and Joyce [47] and the uncorrected one gave almost similarly shaped

reflectance profiles. Therefore, we may conclude that there was a good match between the spectral patterns acquired using those methods in waters exceeding −1.5 m depth.

### 3.3. Benthic Habitat Classification Performance

The spectral fidelity of glint-corrected pixel values was tested based on five habitat-specific classes of corals—TC, BC, PC, R, and S for Site-1. Figures 6 and 7 show benthic habitat maps when the pre-processing techniques of Strategy-1 and Strategy-2 were applied, respectively.

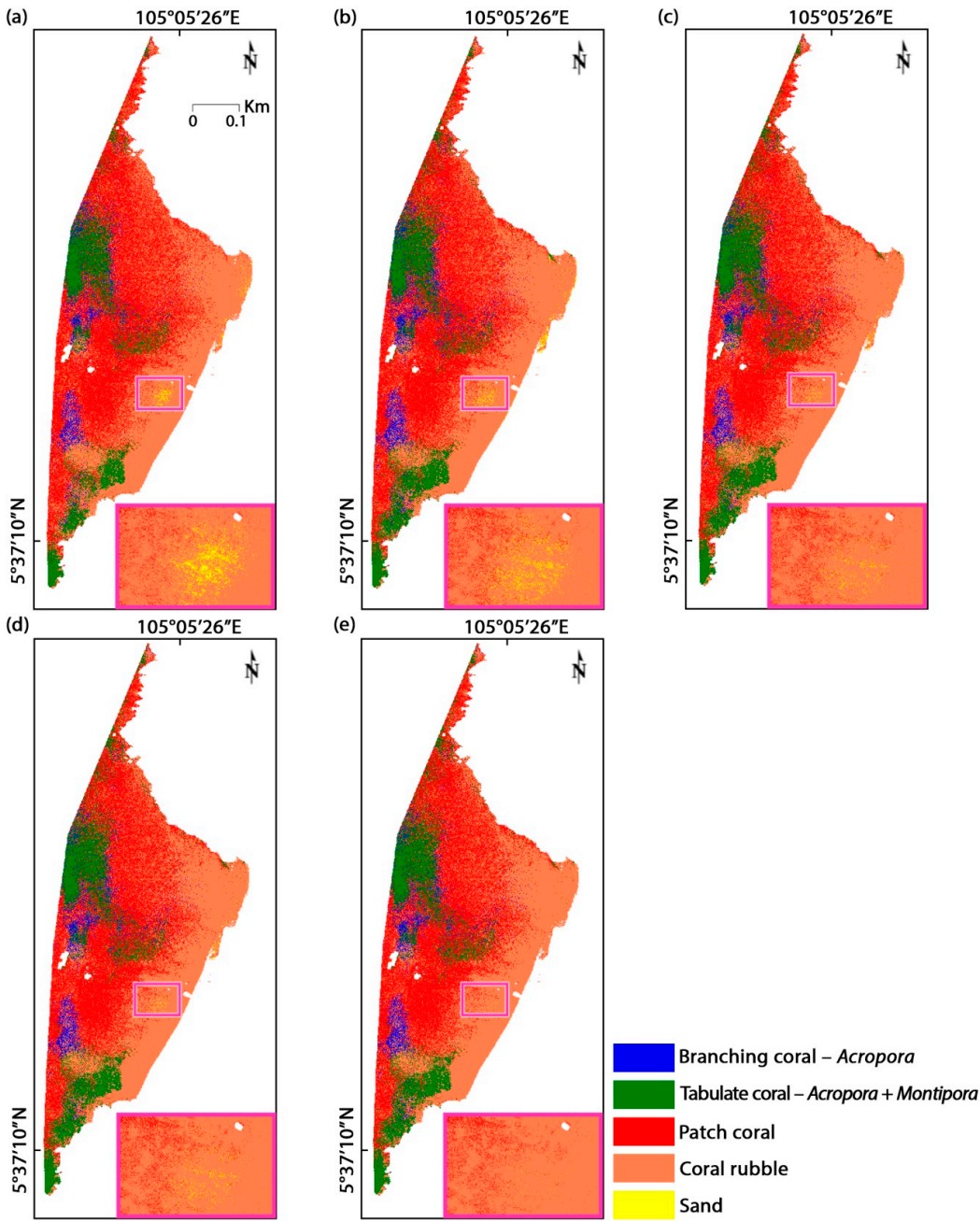

**Figure 6.** Benthic habitat classification maps of Site-1 (Pantai Pasir Cina) produced by supervised support vector machine (SVM) classification of UAV data. Strategy-1 was used to pre-process data. (**a**) Uncorrected image; images corrected using the methods proposed by (**b**) Lyzenga et al. [35], (**c**) Joyce [47], (**d**) Hedley et al. [28], and (**e**) Goodman et al. [29].

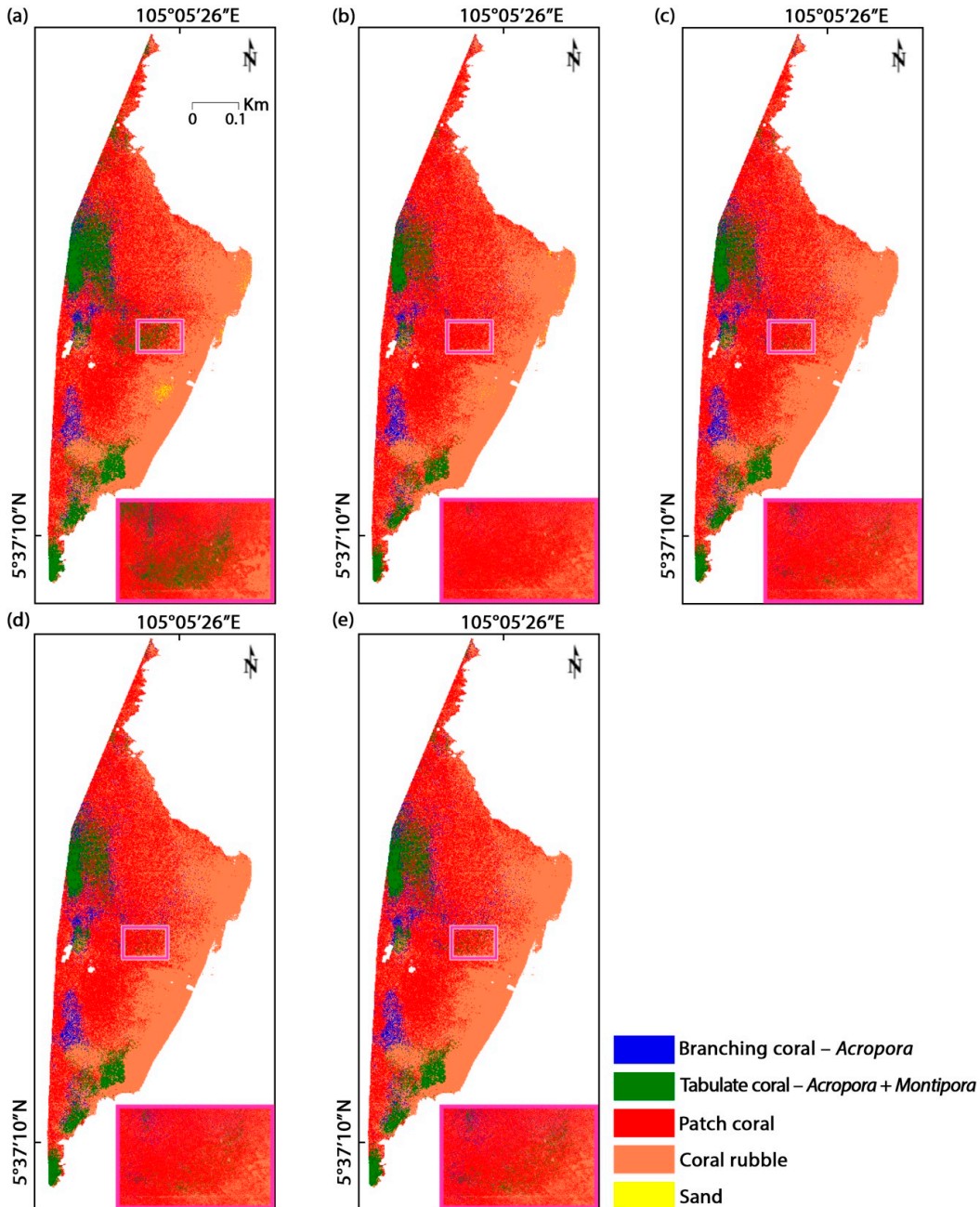

**Figure 7.** Benthic habitat classification maps of Site-1 (Pantai Pasir Cina) produced by supervised SVM classification of UAV data. Strategy-2 was used to pre-process data. (**a**) Uncorrected image; images corrected using the methods proposed by (**b**) Lyzenga et al. [35], (**c**) Joyce [47], (**d**) Hedley et al. [28], and (**e**) Goodman et al. [29].

In general, Site-1 was characterized by near-shore areas dominated by the "R" and "S" classes; the "PC", "TC", and "BC" classes were in the middle, and the "R" and "S" classes were in >−7 m deep water. A comparison of the two strategies showed that, in near-shore areas where "TC" and "S" classes are present, Strategy-2 performed better than Strategy-1. The higher glint-covered areas were misclassified as class "S" (sand) in both uncorrected and inadequately glint-corrected images (zoomed-in regions marked by a red square in Figures 6 and 7). The overestimation of class "S" was obviously evident in images glint-corrected using the method proposed by Lyzenga et al. [35] under Strategy-1 (zoomed-in view in Figure 6b) compared with other techniques. Strategy-2 with the method proposed by Lyzenga et al. [35], however, performed better (Figure 7b), as pixels of class "S" were

correctly identified in the same region. The glint removal procedure suggested by Lyzenga et al. [35] used in this study under Strategy-2 offered satisfactory, comparatively practical results, and provided better-quality coral habitat maps (Figure 7b).

The same pre-processing procedure, Strategy-2, with four glint removal algorithms, was implemented on an external validation site (Site-2) prior to generating benthic habitat maps (Figure 8). The site was dominated by PC in areas of >−7 m water depth, with an assemblage of "TC" and "BC" classes in the middle and an assemblage of "S" and "R" classes in the near-shore areas (eastern part of the site). Figure 8 (in zoomed-in regions) shows obviously that the methods proposed by Lyzenga et al. [35], Hedley et al. [28], and Joyce [47] treated near-shore shallow water pixels around the north side of the site as dominated by "S", whereas the method proposed by Goodman et al. [29] was not disturbed by shallow water depth, and correctly identified those areas as dominated by the "R" class.

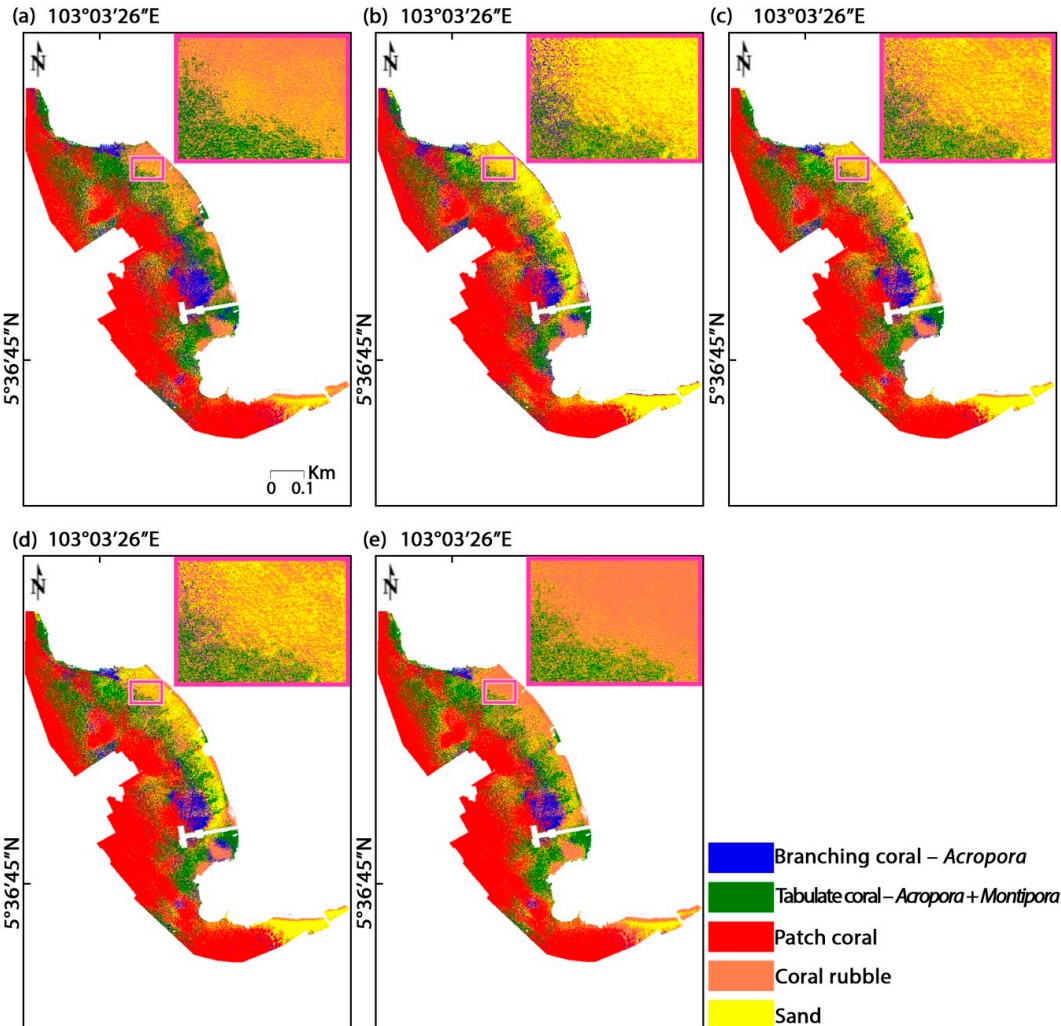

**Figure 8.** Benthic habitat classification maps of Site-2 (Pantai Vietnam) produced by supervised SVM classification of UAV data. Strategy-2 was used to pre-process data. (**a**) Uncorrected image; images corrected using the methods proposed by (**b**) Lyzenga et al. [35], (**c**) Joyce [47], (**d**) Hedley et al. [28], and (**e**) Goodman et al. [29].

The classification maps were further subjected to accuracy assessment using user's accuracy (UA), producer's accuracy (PA), overall accuracy (OA), and Kappa coefficient of agreement ($k$). Table 3 shows that Strategy-2 produced a higher overall classification accuracy compared to Strategy-1 for Site-1. When the glint removal procedures were applied, OA and $k$ increased by more than 13% compared to the uncorrected image. OA and $k$ measures of classification accuracies showed that the

method proposed by Lyzenga et al. [35] under Strategy-2 had the best potential (OA = 87.4%, *k* = 0.831), followed by Strategy-1 with the same glint removal method (OA = 86.0%, *k* = 0.812) and Strategy-2 with Joyce [47] (OA = 85.5%, *k* = 0.805) for discrimination of corals from non-corals.

**Table 3.** Summary of confusion matrices for Site-1 (Pantai Pasir Cina) benthic habitat maps using the sun glint uncorrected dataset, and sun glint-corrected datasets (applying Strategy-1 and Strategy-2). The mapped class abbreviations are as described in Table S2 (Supplementary Materials); BC = branching coral- *Acropora*; TC = tabulate coral – *Acropora* + *Montipora*; PC = patch coral; R = coral rubble; S = sand. UA (%) = user's accuracy as a percentage; PA (%) = producer's accuracy as a percentage; OA = overall accuracy; Correctly mapped = number of correctly mapped classes over the total validation data; *k* = Kappa coefficient of agreement.

| Site-1: Strategy-1 | | | | | | | | | | |
|---|---|---|---|---|---|---|---|---|---|---|
| Mapped class | Uncorrected | | Lyzenga et al. [35] | | Joyce [47] | | Hedley et al. [28] | | Goodman et al. [29] | |
| | UA (%) | PA (%) | UA (%) | PA (%) | UA (%) | PA (%) | UA (%) | PA (%) | UA (%) | PA (%) |
| BC | 70.5 | 70.5 | 90.9 | 82.0 | 90.9 | 82.0 | 90.9 | 82.0 | 92.3% | 78.7 |
| TC | 62.3 | 62.3 | 95.3 | 88.4 | 95.3 | 88.4 | 95.3 | 88.4 | 95.5% | 91.3 |
| PC | 65.0 | 76.1 | 83.9 | 88.9 | 75.7 | 90.6 | 75.5 | 89.7 | 72.9% | 89.7 |
| R | 83.8 | 67.9 | 80.9 | 95.6 | 82.9 | 95.6 | 81.9 | 95.6 | 82.9% | 95.6 |
| S | 55.8 | 64.9 | 100.0 | 43.2 | 100.0 | 10.8 | 100.0 | 8.1 | 100.0% | 2.7 |
| Correctly mapped | 292/421 | | 362/421 | | 352/421 | | 350/421 | | 348/421 | |
| OA | 69.4% | | 86.0% | | 83.6% | | 83.1% | | 82.7% | |
| *k* | 0.600 | | 0.812 | | 0.779 | | 0.772 | | 0.765 | |
| Site-1: Strategy-2 | | | | | | | | | | |
| Mapped class | | | Lyzenga et al. [35] | | Joyce [47] | | Hedley et al. [28] | | Goodman et al. [29] | |
| | | | UA (%) | PA (%) | UA (%) | PA (%) | UA (%) | PA (%) | UA (%) | PA (%) |
| BC | | | 89.7 | 85.2 | 92.6 | 82.0 | 89.3 | 82.0 | 90.3 | 91.8 |
| TC | | | 100.0 | 91.3 | 100.0 | 85.5 | 100.0 | 87.0 | 100.0 | 68.1 |
| PC | | | 91.1 | 87.2 | 84.7 | 89.7 | 86.8 | 89.7 | 82.7 | 98.3 |
| R | | | 78.4 | 97.8 | 77.5 | 95.6 | 74.9 | 95.6 | 76.2 | 95.6 |
| S | | | 100.0 | 45.9 | 100.0 | 40.5 | 100.0 | 24.3 | 100.0 | 2.7 |
| Correctly mapped | | | 368/421 | | 360/421 | | 355/421 | | 350/421 | |
| OA | | | 87.4% | | 85.5% | | 84.3% | | 83.1% | |
| *k* | | | 0.831 | | 0.805 | | 0.788 | | 0.771 | |

To test the robustness of Strategy-2, the benthic habitat classification was applied to an external validation site, i.e., Site-2 (Table 4). The detailed confusion matrices for all glint correction methods for both sites are given in the Supplementary Materials (Tables S3–S16). Strategy-2 improved the classification accuracy from a low of 65.3% OA (*k* = 0.551) in the uncorrected image to 80–87% OA for corrected images—an increase of more than 15%. Table 4 shows that Strategy-2 with glint removal using the method proposed by Lyzenga et al. [35] produced the highest OA and *k* (OA = 86.9%, *k* = 0.827) compared to other methods. The method proposed by Goodman et al. [29] provided the most unsatisfactory benthic habitat classification results compared with all other glint removal techniques for both sites. Therefore, in order to achieve a greater accuracy in coral classifications, glint removal using the method proposed by Lyzenga et al. [35] following Strategy-2 is suggested.

**Table 4.** Summary of confusion matrices for the external validation site (Site-2 Pantai Vietnam) benthic habitat maps using the sun glint uncorrected dataset, and sun glint-corrected datasets (applying Strategy-2). The mapped classes abbreviations are as described in Table S2 (Supplementary Materials); BC = branching coral - *Acropora*; TC = tabulate coral – *Acropora + Montipora*; PC = patch coral; R = coral rubble; S = sand. UA (%) = user's accuracy as a percentage; PA (%) = producer's accuracy as a percentage; OA = overall accuracy; Correctly mapped = number of correctly mapped classes over the total validation data; *k* = Kappa coefficient of agreement.

| | Site-2: Strategy-2 | | | | | | | | | |
|---|---|---|---|---|---|---|---|---|---|---|
| Mapped class | Uncorrected | | Lyzenga et al. [35] | | Joyce [47] | | Hedley et al. [28] | | Goodman et al. [29] | |
| | UA (%) | PA (%) | UA (%) | PA (%) | UA (%) | PA (%) | UA (%) | PA (%) | UA (%) | PA (%) |
| BC | 55.9 | 82.6 | 76.1 | 73.9 | 75.7 | 81.2 | 82.4 | 81.2 | 83.3 | 79.7 |
| TC | 47.9 | 61.6 | 89.6 | 82.2 | 84.2 | 87.7 | 83.1 | 94.5 | 70.8 | 63.0 |
| PC | 76.3 | 71.0 | 92.8 | 90.0 | 91.1 | 92.0 | 90.6 | 87.0 | 83.3 | 80.0 |
| R | 87.0 | 67.6 | 89.7 | 93.9 | 89.8 | 89.2 | 89.1 | 82.4 | 83.5 | 92.6 |
| S | 22.7 | 13.9 | 75.0 | 83.3 | 67.9 | 52.8 | 52.4 | 61.1 | 68.6 | 66.7 |
| Correctly mapped | 278/426 | | 370/426 | | 363/426 | | 356/426 | | 342/426 | |
| OA | 65.3% | | 86.9% | | 85.2% | | 83.6% | | 80.3% | |
| *k* | 0.551 | | 0.827 | | 0.805 | | 0.786 | | 0.739 | |

## 4. Discussion

UAV images was widely used in benthic habitat studies because of their flexible and rapidly affordable platform, with control over flying height and time, making them suitable for mapping coral communities [59]. The effect of sun glint that reduces the number of suitable (useful) images is a concern for satellite imagery, including UAV users [10]. Attempting to minimize sun glint by restricting flying time would obviously limit coral remote sensing because acquisition of satellite images during the lowest low tide time is preferred to avoid the confounding effect of water columns in optical remote sensing [14]. Some suggested a certain degree of tilting to minimize glint, while many satellite platforms such as MODIS (Moderate-Resolution Imaging Spectroradiometer) and MERIS (Medium-Resolution Imaging Spectrometer) do not have such capability [60]. Therefore, instead of avoiding sun glint, it is necessary to investigate how to tackle sun glint problem. Although many glint removal algorithms (Table S1, Supplementary Materials) were suggested for shallow-water remote sensing applications, the level of accuracy in applications for coral mapping still needs to be validated. Unlike other terrestrial mapping applications, submerged aquatic vegetation analyses from satellite imagery are exposed to different confounding factors including water depth and quality [4]. It is crucial to assess the properties of reflectance spectra of glint-removed products and propose suitable techniques for coral habitat mapping, because improving the retrieval of water-leaving radiance determines the level of classification accuracy.

Commonly, Strategy-1 is suggested as a tool to remove glint, which involves reconstructing the whole glint-contaminated image [22]. On the other hand, it is possible to employ glint removal algorithm in glint-contaminated areas only; thus, Strategy-2 would be a better idea than Strategy-1 for classification and distribution mapping of coral habitats. In this context, glint removal using Strategy-2 was demonstrated to be effective in removing surface glint from UAV imagery. The spectral artefacts of high water-leaving reflectance were greatly reduced using the method proposed by Lyzenga et al. [35] (Figure 4b(ii) and Figure 5), providing satisfactory results for both shallow-water (near-shore) and deep-water pixels (Figure 7). Lyzenga et al. [35] reported that their method could generate better-quality images where water is shallow and vegetation is emerging. Similarly, this study showed that glint-corrected images using the method proposed by Lyzenga et al. [35] can yield more accurate maps of coral distribution than the other three methods of glint correction.

With respect to overcorrection, all methods except for that proposed by Lyzenga et al. [35] showed a tendency to overestimate class "S" coverage, especially in the shallow near-shore areas (Figures 6 and 7). The major weakness of currently used glint removal procedures in some circumstances and the causes of overcorrection were previously documented [27,28]. Once submerged vegetation such as seagrass reaches the water surface during low tide, cyanobacteria in the water or the existence of scum may cause overcorrection due to high reflectance spectra in NIR and short-wave infrared (SWIR) regions. This violates the zero NIR (and SWIR) assumption in the glint correction algorithm. In the case of this study, potential sources of overcorrection could be the coincidental existence of class "S" (and class "R") and sun glint events in shallow areas, as these substrates often reach near to or above the water surface.

Kay et al. [22] suggested that the relative magnitude and the shape of reflectance profiles are more crucial than the absolute radiometric accuracy. Glint and noise can both change the spectral signature and lead to image misclassification. The best glint correction method should reduce the noise to a minimum, as well as the glint. Strategy-1 exhibited nearly flat spectral profiles, resulting in a reduced performance in the classification of benthic habitats. The lowest reflectance spectra using the method proposed by Lyzenga et al. [35] with Strategy-2 (Figure 5b) illustrated adequate glint correction. Therefore, image pre-processing using Strategy-2 and the glint correction method suggested by Lyzenga et al. [35] could be considered as a robust approach prior to implementing image classification techniques from the spectral analysis point of view.

## 5. Conclusions

Sun glint correction is an essential image pre-processing step of satellite image analysis, which enables improved retrieval of water-leaving radiance in the case of aquatic remote sensing. A comparison of four glint correction algorithms when applied to high-resolution UAV imagery following whole-image (Strategy-1) and glint-impacted (Strategy-2) strategies showed that images pre-processed with Strategy-2 using the method proposed by Lyzenga et al. [35] performed better than the other methods, resulting in reliable maps of corals. The proposed glint removal approach was effective in a limited complex coral environment where there was a great possibility of overestimation of underwater substrates, particularly for class "S". Near-shore areas, including mudflats, contain ecologically important coral communities; the suggested glint removal method can allow the separation of pixels with and without class "S" in the de-glinted images. Preserving underwater reflectance from overcorrection after glint correction will allow studying spatiotemporal changes in water quality. In order to increase the applicability of glint-contaminated imagery, further research is needed to test glint removal methods with combinations of UAV images acquired during different tide conditions and seasons.

**Supplementary Materials:** The following are available online at http://www.mdpi.com/2072-4292/11/20/2422/s1: Table S1: Comparison of different methods used in sun glint correction for coastal and shallow-water mapping; Table S2: In situ data of coral benthic communities used for training and validation of classification, expressed as the number of pixels in the UAV images; Table S3: Error matrix for Site-1 Strategy-1 sun glint uncorrected image for benthic habitat classification. Table S4: Error matrix for Site-1 Strategy-1 sun glint-corrected image using the method proposed by Lyzenga et al. [35] for benthic habitat classification. Table S5: Error matrix for Site-1 Strategy-1 sun glint-corrected image using the method proposed by Joyce [47] for benthic habitat classification. Table S6: Error matrix for Site-1 Strategy-1 sun glint-corrected image using the method proposed by Hedley et al. [28] for benthic habitat classification. Table S7: Error matrix for Site-1 Strategy-1 sun glint-corrected image using the method proposed by Goodman et al. [29] for benthic habitat classification. Table S8: Error matrix for Site-1 Strategy-2 sun glint-corrected image using the method proposed by Lyzenga et al. [35] for benthic habitat classification. Table S9: Error matrix for Site-1 Strategy-2 sun glint-corrected image using the method proposed by Joyce [47] for benthic habitat classification. Table S10: Error matrix for Site-1 Strategy-2 sun glint-corrected image using the method proposed by Hedley et al. [28] for benthic habitat classification. Table S11: Error matrix for Site-1 Strategy-2 sun glint-corrected image using the method proposed by Goodman et al. [29] for benthic habitat classification. Table S12: Error matrix for Site-2 Strategy-2 sun glint uncorrected image for benthic habitat classification. Table S13: Error matrix for Site-2 Strategy-2 sun glint-corrected image using the method proposed by Lyzenga et al. [35] for benthic habitat classification. Table S14: Error matrix for Site-2 Strategy-2 sun glint-corrected image using the method proposed by Joyce [47] for benthic habitat classification. Table S15: Error matrix for Site-2 Strategy-2 sun glint-corrected image using the method proposed by Hedley et al. [28] for benthic habitat classification. Table S16: Error matrix for Site-2 Strategy-2 sun glint-corrected image using the method proposed by Goodman et al. [29] for benthic habitat classification.

**Author Contributions:** Conceptualization, W.S.C. and M.S.H.; methodology, W.S.C. and M.S.H.; software, A.M.M; validation, A.M.M., W.S.C., and I.K.; formal analysis, W.S.C., C.D.M.S., and I.K.; investigation, W.S.C. and C.D.M.S.; resources, W.S.C.; data curation, W.S.C. and C.D.M.S.; writing—original draft preparation, W.S.C.; writing—review and editing, A.M.M. and M.S.H.; visualization, W.S.C.; supervision, A.M.M.; project administration, A.M.M.; funding acquisition, A.M.M.

**Funding:** This research was supported by the FRGS (Project Code: FRGS/1/2019/WAB05/UMT/02/3 and Vot number: 59562) from the Ministry of Higher Education, Malaysia.

**Acknowledgments:** This research was supported by the FRGS (Project Code: FRGS/1/2019/WAB05/UMT/02/3 and Vot number: 59562) from the Ministry of Higher Education, Malaysia. We would like to thank the Academic Editor and three anonymous reviewers whose comments helped to improve the manuscript from its original version.

**Conflicts of Interest:** The authors report no potential conflicts of interest.

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
