# Peer review of "Coral Reef Mapping of UAV: A Comparison of Sun Glint Correction Methods"

_remotesensing, doi:10.3390/rs11202422_

Round 1

Reviewer 1 Report

This paper is a comparison of sun glint correction methods applied to UAV data over coral reef areas. The introduction tries to argue that methods for airborne data and over coral reefs have not been developed very much, which isn't true, in fact many of the citations are for airborne and/or coral reef areas. It's true that there aren't many papers for drone camera imagery, but this paper doesn't really highlight any specific aspects of this kind of data (it could, and that would strengthen the paper). The main novel aspect of the paper appears to be the method for separating no-glint and glint areas, but this is explained in only one sentence and is not at all clear. Much more detail on this is needed.

One issue is that the method described as "Kutser et al. 2009", is not an implementation of that method. That method requires narrow bands to identify the oxygen absorption feature at 760nm, I doubt the bands of the sensor used here can be sufficiently narrow (no information given). In addition the reference wavelength is changed to 668nm from 760nm, there is an oxygen absorption feature at 686nm (not 668nm) but it is much weaker than at 760nm, it can be seen in the plots in Kutser et al. It's not clear if it's intended to target this feature. In addition the 560nm band is used which is well into the green and will be effected by subsurface reflectance. In general the relationship between 560nm, 668nm and 840nm will be massively effected by the subsurface reflectance, this may explain the subsequent performance in the classification, but doesn't make it a good glint removal procedure. In Kutser et al. all bands used were in the NIR.

While assessing the results in terms of benthic classification is useful as the final aim, the best assessment (in the absence of in-situ reflectance data) is the profile of reflectance in each band across a transect containing glint. The glint peaks should be clear, and these should be smoothed out in the corrected images, see plots in Kay et al. 2009 for example. However in this paper these plots are presented very badly and it is hard to see anything.

Finally I don't understand why there is a big jump in the corrected reflectance between no-glint and glint areas (transects of Fig 4). All of these methods should just be removing glint, the glint peaks should be smoothed out and this should be the main visible result. Again, see Fig 8 in Kay et al. 2009. A small step in overall reflectance might be expected dependent on the method but not the huge changes seen in Fig 4. Glint is supposed to be removed, so why does reflectance increase? I question if there isn't something fundamentally wrong with the method of application.

In summary I think a paper like this could be publishable and useful, but substantial work is required.
1) Explain better the method for identifying glint and no-glint areas, this is potentially the most useful part of the paper, given that the other methods used here are already published.
2) Ensure methods are applied correctly, explain the big change in reflectance, and if it's real find some examples where it doesn't occur, because it's not usual.
3) The "Kutser" method can't be called that, so either remove it or give it a new name, although methods shouldn't be just made up without a physical justification.
4) Present results better, especially transects of reflectance.

Specific comments:

Table 1. I would remove all references to "ocean color" methods since they are relevant in this paper. It would be enough to mention they exist and why they aren't applicable.

Line 65. It's not really true to say that the images don't contain this information, the issue is that the information is not readily extractable by methods such as classification etc, because of the sun glint. Glint correction does not add information it just reorganizes the information to make it more amenable to subsequent analysis methods. The information was there all the time.

Paragraph at 107. It is quite inaccurate to give the impression existing methods a have been primarily developed for satellite data and also not applied in coral reef areas. Probably at least half the cited papers are using airborne imagery or are on coral reefs. Many of the authors cited work primarily on coral reefs, Hochberg, Hedley, Joyce, Goodman etc.

Line 124, not necessarily zero, it depends on what is taken as the NIR baseline. This is the main (only) difference between several methods and is covered in Kay et al. The NIR reflectance may not be expected to be zero because of aerosols in the atmosphere, this is dependent on if atmospheric correction has been applied.

Line 127, as above, zero NIR is not an assumption in many common methods. What is an assumption is spatially constant NIR.

Line 129, which methods appropriate for high spatial resolution imagery require these? I don't think it is common, I suggest to cite the papers.

Line 139, I would rephrase this sentence because to many readers it's common practice to apply glint removal to airborne imagery of coral reefs, and has been for probably 20 years. That doesn't mean a new paper on it can't be a useful contribution though.

Line 269, I think the point of this method is that it is NOT sensitive to outliers. This sentence seems to say the opposite.

Section 2.2.2 isn't this calculation mathematically the same as a least-squares regression? Please clarify if it is just the same calculation or different in some way.

Line 298 - "mainly differ" or ONLY differ? Why not simplify this by just showing one equation and stating that the subtracted value can take these three options? It would be much clearer for the reader. I think this point is made quite well in Kay et al. Also, how is the mode calculated? Literally the most common value, or through the fit to a distribution?

Line 319 - As discussed above I don't think this can be called the Kutser et al. method because it lacks the key features of that method.

Line 338 - the key aspect of strategy 2, the identification of glint areas, is explained in only one sentence. It is not at all clear how this is done, yet it's likely the main novel contribution of the paper. Since glint varies from pixel to pixel, how can glint contaminated regions be identified? Much more explanation and assessment of this aspect is needed.

Line 414 - how is this normalisation done? What does it mean? the pixel values are rescaled so the brightest is 1? what happens to pixels that are less than zero (which can happen with some deglint methods) or greater than 1? All the images should be subjected to the same contrast stretch, so the visual appearance shows the difference between them, not individually normalised.

Fig 3/4. I wouldn't show such large scale images as its very hard to see what things are. Concentrate on one or more zoomed in regions where the glint can be seen. Also its necessary to show where the training areas for the glint corrections are as this may effect the results of specific methods. Since, for example, the subtracted NIR value is derived from these areas.

Fig 3/4. The transect plots should be rescaled so that features can be seen, In figure 3 they just look like horizontal lines. Why is there a big discontinuity seen in Fig 4 between the no glint and glint areas? This should not occur for most methods.

Fig 5. Very hard to see which line is which, some colors are similar, symbols all just look like tiny dots, they key contains six lines but only five can be seen. Label the plots directly to say what they represent.

Author Response

Reviewer 1

This paper is a comparison of sun glint correction methods applied to UAV data over coral reef areas. The introduction tries to argue that methods for airborne data and over coral reefs have not been developed very much, which isn't true, in fact many of the citations are for airborne and/or coral reef areas. It's true that there aren't many papers for drone camera imagery, but this paper doesn't really highlight any specific aspects of this kind of data (it could, and that would strengthen the paper). The main novel aspect of the paper appears to be the method for separating no-glint and glint areas, but this is explained in only one sentence and is not at all clear. Much more detail on this is needed.

RE: Yes it has pointed out in the Introduction section. The added sentence is ‘There are few studies of sun glint correction methods applied to UAV camera imagery across coral reef areas.’

One issue is that the method described as "Kutser et al. 2009", is not an implementation of that method. That method requires narrow bands to identify the oxygen absorption feature at 760nm, I doubt the bands of the sensor used here can be sufficiently narrow (no information given). In addition the reference wavelength is changed to 668nm from 760nm, there is an oxygen absorption feature at 686nm (not 668nm) but it is much weaker than at 760nm, it can be seen in the plots in Kutser et al. It's not clear if it's intended to target this feature. In addition the 560nm band is used which is well into the green and will be effected by subsurface reflectance. In general the relationship between 560nm, 668nm and 840nm will be massively effected by the subsurface reflectance, this may explain the subsequent performance in the classification, but doesn't make it a good glint removal procedure. In Kutser et al. all bands used were in the NIR.

RE: The glint correction method ‘Kutser et al. 2009’ removed as suggested. Revised texts in results and discussion sections. Accordingly, edited Figures 2-8 and all tables.

While assessing the results in terms of benthic classification is useful as the final aim, the best assessment (in the absence of in-situ reflectance data) is the profile of reflectance in each band across a transect containing glint. The glint peaks should be clear, and these should be smoothed out in the corrected images, see plots in Kay et al. 2009 for example. However in this paper these plots are presented very badly and it is hard to see anything.

RE: Rescaled all plots.

Finally I don't understand why there is a big jump in the corrected reflectance between no-glint and glint areas (transects of Fig 4). All of these methods should just be removing glint, the glint peaks should be smoothed out and this should be the main visible result. Again, see Fig 8 in Kay et al. 2009. A small step in overall reflectance might be expected dependent on the method but not the huge changes seen in Fig 4. Glint is supposed to be removed, so why does reflectance increase? I question if there isn't something fundamentally wrong with the method of application.

RE: This is due to transition between glint-corrected and uncorrected images when implemented under Strategy-2 and it also explained inside the manuscript. Previous studies including Kay et al. 2009 did not apply Strategy-2 and therefore, results are not comparable.

In summary I think a paper like this could be publishable and useful, but substantial work is required.

1) Explain better the method for identifying glint and no-glint areas, this is potentially the most useful part of the paper, given that the other methods used here are already published.

2) Ensure methods are applied correctly, explain the big change in reflectance, and if it's real find some examples where it doesn't occur, because it's not usual.

3) The "Kutser" method can't be called that, so either remove it or give it a new name, although methods shouldn't be just made up without a physical justification.

4) Present results better, especially transects of reflectance.

RE: All comments considered and much effort was given to comply with them. Removed the The "Kutser" method as suggested.

Specific comments:

Table 1. I would remove all references to "ocean color" methods since they are relevant in this paper. It would be enough to mention they exist and why they aren't applicable.

RE: Description on the ‘ocean color’ methods have been removed from Table 1 as suggested.

Line 65. It's not really true to say that the images don't contain this information, the issue is that the information is not readily extractable by methods such as classification etc, because of the sun glint. Glint correction does not add information it just reorganizes the information to make it more amenable to subsequent analysis methods. The information was there all the time.

RE: The revised sentence now reads as ‘Due to sun glint, useful information for water quality analysis and benthic feature extraction cannot be readily extracted by techniques such as image classification.’

Paragraph at 107. It is quite inaccurate to give the impression existing methods a have been primarily developed for satellite data and also not applied in coral reef areas. Probably at least half the cited papers are using airborne imagery or are on coral reefs. Many of the authors cited work primarily on coral reefs, Hochberg, Hedley, Joyce, Goodman etc.

RE: The following sentence revised: ‘There are few studies of sun glint correction methods applied to UAV camera imagery across coral reef areas.’

Line 124, not necessarily zero, it depends on what is taken as the NIR baseline. This is the main (only) difference between several methods and is covered in Kay et al. The NIR reflectance may not be expected to be zero because of aerosols in the atmosphere, this is dependent on if atmospheric correction has been applied.

RE: The sentence removed.

Line 127, as above, zero NIR is not an assumption in many common methods. What is an assumption is spatially constant NIR.

RE: The revised sentence now reads as: ‘However, due to water quality (turbidity), the basic assumption – spatially constant NIR - may not hold.’

Line 129, which methods appropriate for high spatial resolution imagery require these? I don't think it is common, I suggest to cite the papers.

RE: Cited relevant papers.

Line 139, I would rephrase this sentence because to many readers it's common practice to apply glint removal to airborne imagery of coral reefs, and has been for probably 20 years. That doesn't mean a new paper on it can't be a useful contribution though.

RE: The rephrased sentence is ‘An appropriate sun glint removal technique has not yet been examined to improve UAV imagery and the usefulness of this method for coral and related habitat classification and distribution mapping.’

Line 269, I think the point of this method is that it is NOT sensitive to outliers. This sentence seems to say the opposite.

RE: Corrected.

Section 2.2.2 isn't this calculation mathematically the same as a least-squares regression? Please clarify if it is just the same calculation or different in some way.

RE: the revised sentence now reads as: ‘We computed the covariance of each visible band () relative to the NIR band () (as expressed in Equation 2) to implement glint removal method proposed by Lyzenga et al. [50]’

Line 298 - "mainly differ" or ONLY differ? Why not simplify this by just showing one equation and stating that the subtracted value can take these three options? It would be much clearer for the reader. I think this point is made quite well in Kay et al. Also, how is the mode calculated? Literally the most common value, or through the fit to a distribution?

RE: Removed subsections and reduced the length of the section by summarizing the tested methods as suggested by Reviewer #3.

Line 319 - As discussed above I don't think this can be called the Kutser et al. method because it lacks the key features of that method.

RE: As mentioned earlier, removed the Kutser et al. method.

Line 338 - the key aspect of strategy 2, the identification of glint areas, is explained in only one sentence. It is not at all clear how this is done, yet it's likely the main novel contribution of the paper. Since glint varies from pixel to pixel, how can glint contaminated regions be identified? Much more explanation and assessment of this aspect is needed.

RE: Reference added [13,35] to target readers that are interested in the rule-based (decision tree) approach for comprehensive study.

Line 414 - how is this normalisation done? What does it mean? the pixel values are rescaled so the brightest is 1? what happens to pixels that are less than zero (which can happen with some deglint methods) or greater than 1? All the images should be subjected to the same contrast stretch, so the visual appearance shows the difference between them, not individually normalised.

RE: sufficient information pertaining this issue is given in the methods section.

Fig 3/4. I wouldn't show such large scale images as its very hard to see what things are. Concentrate on one or more zoomed in regions where the glint can be seen. Also its necessary to show where the training areas for the glint corrections are as this may effect the results of specific methods. Since, for example, the subtracted NIR value is derived from these areas.

RE: revised figures 3 and 4 as suggested.

Fig 3/4. The transect plots should be rescaled so that features can be seen, In figure 3 they just look like horizontal lines. Why is there a big discontinuity seen in Fig 4 between the no glint and glint areas? This should not occur for most methods.

RE: same as above.

Fig 5. Very hard to see which line is which, some colors are similar, symbols all just look like tiny dots, they key contains six lines but only five can be seen. Label the plots directly to say what they represent.

RE: As suggested by the reviewer, we have revised Figure 5.

Reviewer 2 Report

The authors compare multiple methods of glint correction for multispectral, high spatial resolution imagery collected over coral reefs.

This is an interesting and potentially useful study. Overall the manuscript is well written. However, there are numerous caveats and assumptions inherent to the methodology which must be explicitly mentioned.

In general, the methods examined rely on homogenous water column optical properties across a study area. This is likely not the case over coral reef environments, especially if deep water pixels are utilized in obtaining the glint offsets (cite Russell et al. 2019 Water Column Optical Properties of Pacific Coral Reefs Across Geomorphic Zones and in Comparison to Offshore Waters), and a discussion of this point or at least acknowledgment is needed.

ln 19: streategy to strategy

Table 1: this is an impressive and very useful summary

ln 121: Add citation Fogarty et al. 2018 The influence of a sandy substrate, seagrass, or highly turbid water on Albedo and surface heat flux

ln 159 – 161: the English in these sentences needs to be fixed.

Section 2.1: what is the ground sampling distance/pixel size of the imagery?

ln 182: “lousy weather” is not an appropriate descriptive term.

ln 233: if it’s a lensed system it measures radiance, not irradiance

ln 237: need more details about both the reflectance panel utilized (including size and how many pixels it is in an image) and information about the downwelling light sensor and how fluctuating light levels during a flight was compensated for

ln 243: how was the radiometric calibration performed?

ln 271: a bit to “was”

Sec 2.2.5 – I question if this band substitution is valid – what is the band width of this sensor? These bands are very far spectrally from those used in Kutser.

ln 452: what do you mean by field spectra?

ln 457: I do not believe this assumption is valid. The benthic reflectance is not necessarily constant over this transect and the change cannot be attributed only to glint.

ln 472: The Kutser method produced a lower value than the others, but without independent or in-situ verification it can’t be considered closer to any “true” value, unless I’m missing something here?

Figure 5: switch the R and B positions along the x axis. convention is to have shorter wavelengths to the left.

There is an issue in how benthic classification is performed here. I do not believe that there is sufficient information in the reflectance data to distinguish all of these classes. In particular, I do not believe that species of coral or coral communities could be separated (a challenge even with hyperspectral data). Or that sand and coral rubble could be separated, or that coral could be distinguished from other submerged photoautotrophs like seagrass or algae.

This study is useful for shallow coastal and aquatic environments beyond just coral reefs, but the authors make no mention of this. At least adding a short discussion of improved retrievals for other applications like submerged vegetation mapping would help make the case for the utility of this study and the correction methodology presented.

Author Response

Reviewer 2

The authors compare multiple methods of glint correction for multispectral, high spatial resolution imagery collected over coral reefs.

This is an interesting and potentially useful study. Overall the manuscript is well written. However, there are numerous caveats and assumptions inherent to the methodology which must be explicitly mentioned.

In general, the methods examined rely on homogenous water column optical properties across a study area. This is likely not the case over coral reef environments, especially if deep water pixels are utilized in obtaining the glint offsets (cite Russell et al. 2019 Water Column Optical Properties of Pacific Coral Reefs Across Geomorphic Zones and in Comparison to Offshore Waters), and a discussion of this point or at least acknowledgment is needed.

RE: Referred literature cited.

ln 19: streategy to strategy

RE: Corrected.

Table 1: this is an impressive and very useful summary

RE: Thank you.

ln 121: Add citation Fogarty et al. 2018 The influence of a sandy substrate, seagrass, or highly turbid water on Albedo and surface heat flux

RE: Referred literature cited.

ln 159 – 161: the English in these sentences needs to be fixed.

RE: Revised the sentence – ‘Note that data acquired from Site-1 was used to evaluate sun glint models, while data acquired from Site-2 (twice as large as Site-1) was used to further validate the Site-1 method.’

Section 2.1: what is the ground sampling distance/pixel size of the imagery?

RE: These information are available in lines 243-246.

ln 182: “lousy weather” is not an appropriate descriptive term.

RE: revised the sentence ‘This design allowed easy retrieval of drones when the weather was unexpected, marine bird interference or system failure.’

ln 233: if it’s a lensed system it measures radiance, not irradiance

RE: Corrected.

ln 237: need more details about both the reflectance panel utilized (including size and how many pixels it is in an image) and information about the downwelling light sensor and how fluctuating light levels during a flight was compensated for

RE: The following new sentences added in the method section describing the reflectance panel: The sky was well-lit with low complete cloud cover during the UAV data acquisition (see Table 3). A suggested in the technical notes of the sensor supplier, the calibration panel was used for radiometric calibration during the sunny day (see https://support.micasense.com/hc/en-us/articles/360025336894). The calibration panel has a visible and near-infrared spectrum calibration curve. The sensor supplier provides the calibration data in the range of 400 nm to 850 nm as absolute reflectance (a value between 0 and 1). In this study, the average panel albedo for band red = 0.68, green = 0.69, blue = 0.68, NIR = 0.63, red edge = 0.68 was used represent the calibration curve by five reflectance values or albedos, one for each of the five bands of the Micasense RedEdge camera. The panel dimension is 15.5 cm by 15.5 cm and for radiometric calibration it was ensured that at least one-third of the image width was taken. Due to its most effective in overcast, completely cloudy conditions, the downwelling light sensor data was not used for image post-processing. The drone GPS module was installed at a higher position than the light sensor, thus causing its shadow be cast on the light sensor when the drone changed its cruising direction.

ln 243: how was the radiometric calibration performed?

RE: The following new sentences added in the method section detailing the radiometric correction method: The radiometric calibration was done automatically in the “DSM, Orthomosaic and Index” module of Pix4D Mapper Pro and the radiometric correction type was selected “camera only”. The corresponding calibration panel image was imported into the module for the selected band, an ROI was drawn on the image to define the area of radiometric calibration, and the albedo value was inserted for the selected band. This procedure has been repeated for each of the five bands. The module utilizes the values of some parameters in the EXIF metadata of the drone images to correct variables such as incoming sunlight irradiance, ISO, aperture, shutter speed, vignetting, sensor response and optical system when producing the reflectance map. Further details on the radiometric calibration for Micasense RedEdge camera can be referred to https://support.micasense.com/hc/en-us/articles/115000831714-How-to-Process-RedEdge-Data-in-Pix4D.

ln 271: a bit to “was”

RE: the sentence revised as ‘The original approach has been slightly modified for this study by measuring the glint intensity from a large number of pixels instead of two pixels.’

Sec 2.2.5 – I question if this band substitution is valid – what is the band width of this sensor? These bands are very far spectrally from those used in Kutser.

RE: that’s why Kutser method was removed which was also suggested by Reviewer #1.

ln 452: what do you mean by field spectra?

RE: Corrected; it should be ‘benthic’ spectra.

ln 457: I do not believe this assumption is valid. The benthic reflectance is not necessarily constant over this transect and the change cannot be attributed only to glint.

RE: the word ‘mainly’ added to the sentence.

ln 472: The Kutser method produced a lower value than the others, but without independent or in-situ verification it can’t be considered closer to any “true” value, unless I’m missing something here?

RE: The Kutser method has been removed.

Figure 5: switch the R and B positions along the x axis. convention is to have shorter wavelengths to the left.

RE: revised Figure 5 as suggested.

There is an issue in how benthic classification is performed here. I do not believe that there is sufficient information in the reflectance data to distinguish all of these classes. In particular, I do not believe that species of coral or coral communities could be separated (a challenge even with hyperspectral data). Or that sand and coral rubble could be separated, or that coral could be distinguished from other submerged photoautotrophs like seagrass or algae.

RE: that’s why most popularly used SVM classification method applied to glint corrected images.

This study is useful for shallow coastal and aquatic environments beyond just coral reefs, but the authors make no mention of this. At least adding a short discussion of improved retrievals for other applications like submerged vegetation mapping would help make the case for the utility of this study and the correction methodology presented.

RE: the following sentence added in the Conclusion section: The glint removal approach tested and utilized for this study has potential to map and monitor submerged vegetation such as seagrasses, sea weed or macro algae in the coastal and shallow water environments.

Reviewer 3 Report

The manuscript describes the application of some strategies and correction methods to UAV images, with the aim to reduce and remove the sun glint. The final aim is that of recognizing the coral reef habitats and properly mapping the nearshore areas. Such procedures are applied to two different sites in the Peninsular Malaysia.

The work is interesting and well organized, with clear description of the used techniques. However, the manuscript is too long and require an extensive editing to reach readability.

English use should be carefully checked before resubmission throughout the text: only some specific points are listed in the following.

Major points

As already said, the major problem is the paper length. This should be significantly shortened, starting from the Introduction. Specifically, the text in this section should be summarized a little bit. Further, Table 1 (which takes 6 pages!) needs to be either removed (with the content briefly summarized in the text) or recalled as a supplementary material. Table 1 content is something already known, hence taking all such space is not worth.

In addition, in the end of the Introduction, the novelty of the paper should be properly highlighted. As far as I can understand, the applied approaches are all well-known, hence the only novelty is the application to UAV images.

Within Section 2, I believe that 2.1 and 2.2 could be significantly reduced, removing useless parts and summarizing others which are too verbose. For instance, the description of the correction methods (L268-L330) should be significantly summarized, as these are already known, and put under the same Section 2.2, without use of subsections 2.2.1 to 2.2.5.

I also suggest Table 4 be either moved to supplementary materials or removed and briefly summarized within the text.

To improve figure quality and highlight the differences among the presented results, the image without glint correction should be included in Figure 4 (becoming panel “a”), similarly to what done in Figure 3.

In agreement with Figure 6, the uncorrected image should also be added in Figure 7.

The comparison between the presented results and those related to other sources of video imagery (e.g. satellite images), already presented in the recent literature, could be worth of investigation and should be included in the Discussion section.

Specific points

L120-121: the sentence here is not sufficiently clear, please reword. L125: it should be “may not be valid”. L127: “and necessitate”. L151: the text “(=island)” can be removed. L201: “were in northward direction”. L411: “were quantitatively estimated”. L430: “this method could be useful”. L439: “Presence of … demonstrates”. L481-482: this sentence should be reworded. L498-504: these sentences should be reworded. L557-559: what do the authors mean with “may glint removal …” and with “applications yet to be validated”; I suggest rephrasing. L567: what do the authors mean with “distribution mapping is wide idea”? L575: “overcorrection, all methods”. L583-584: to be reworded. L588: “exhibits … leads”. L599-600: “performs better … and results in reliable”.

Author Response

Reviewer 3

The manuscript describes the application of some strategies and correction methods to UAV images, with the aim to reduce and remove the sun glint. The final aim is that of recognizing the coral reef habitats and properly mapping the nearshore areas. Such procedures are applied to two different sites in the Peninsular Malaysia.

The work is interesting and well organized, with clear description of the used techniques. However, the manuscript is too long and require an extensive editing to reach readability.

English use should be carefully checked before resubmission throughout the text: only some specific points are listed in the following.

Major points

As already said, the major problem is the paper length. This should be significantly shortened, starting from the Introduction. Specifically, the text in this section should be summarized a little bit. Further, Table 1 (which takes 6 pages!) needs to be either removed (with the content briefly summarized in the text) or recalled as a supplementary material. Table 1 content is something already known, hence taking all such space is not worth.

RE: Description on the ‘ocean color’ methods have been removed from Table 1 as suggested.

In addition, in the end of the Introduction, the novelty of the paper should be properly highlighted. As far as I can understand, the applied approaches are all well-known, hence the only novelty is the application to UAV images.

RE: the end paragraph now reads as: ‘Applications involving the use of UAV camera image pre-processing techniques for sun glint effects in coral reef mapping were not extensively evaluated and implemented. An appropriate sun glint removal technique has not yet been examined to improve UAV imagery and the usefulness of this method for coral and related habitat classification and distribution mapping.’

Within Section 2, I believe that 2.1 and 2.2 could be significantly reduced, removing useless parts and summarizing others which are too verbose. For instance, the description of the correction methods (L268-L330) should be significantly summarized, as these are already known, and put under the same Section 2.2, without use of subsections 2.2.1 to 2.2.5.

RE: Removed subsections and reduced the length of the section by summarizing the tested methods.

I also suggest Table 4 be either moved to supplementary materials or removed and briefly summarized within the text.

RE: Table 4 moved to supplementary material.

To improve figure quality and highlight the differences among the presented results, the image without glint correction should be included in Figure 4 (becoming panel “a”), similarly to what done in Figure 3.

RE: revised Figure 4 accordingly.

In agreement with Figure 6, the uncorrected image should also be added in Figure 7.

RE: revised Figure 7 accordingly.

The comparison between the presented results and those related to other sources of video imagery (e.g. satellite images), already presented in the recent literature, could be worth of investigation and should be included in the Discussion section.

Specific points

L120-121: the sentence here is not sufficiently clear, please reword.

RE: Reworded. The texts now read as ‘When sun glint correction is carried out, as mentioned earlier, it is assumed that water leaving signal is nearly zero in the NIR spectrum part, while it could be strong because of the water quality is optically shallow and existence of underwater substrates, for example seagrasses [16] and corals [50,51].’

L125: it should be “may not be valid”.

RE: Corrected.

L127: “and necessitate”.

RE: Corrected.

L151: the text “(=island)” can be removed.

RE: the text removed as suggested. L151

L201: “were in northward direction”.

RE: Corrected.

L411: “were quantitatively estimated”.

RE: replaced the word ‘evaluate’ with ‘estimated’.

L430: “this method could be useful”.

RE: Corrected.

L439: “Presence of … demonstrates”.

RE: Corrected.

L481-482: this sentence should be reworded.

RE: Reworded. The sentence now reads as ‘The near-one reflectance spectrum is an indication of overcorrection.’

L498-504: these sentences should be reworded.

RE: Reworded; sentences now read as ‘The overestimation of the class ‘S’ is obviously evident if glint corrected by Lyzenga et al. (2006) under Strategy-1 (zoomed-in view in Figure 6(b)) compared with other techniques. Strategy-2 with Lyzenga et al. (2006), however, performed better (Figure 7(a)), pixels of the class ‘S’ correctly identified in the same region. Glint removal procedure suggested by Lyzenga et al. (2006) but proposed in this study under Strategy-2 offered satisfactory, comparatively practical results, and provided better quality coral habitat maps (Figure 7(b)).’

L557-559: what do the authors mean with “may glint removal …” and with “applications yet to be validated”; I suggest rephrasing.

RE: the rephrased sentence now reads as ‘Although many glint removal algorithms (Table 1) have been suggested, the level of accuracy in applications for coral mapping still to be validated.’

L567: what do the authors mean with “distribution mapping is wide idea”?

RE: the revised sentence now reads as ‘While glint removal algorithm is possible to employ in the glint-contaminated areas only, investigation on glint removal methods with Strategy-2 would be better idea than applying glint removal to the whole scene (Strategy-1) for coral habitat classification and distribution mapping.’

L575: “overcorrection, all methods”.

RE: removed the word ‘the’.

L583-584: to be reworded.

RE: Reworded. The revised sentence now reads as ‘In the case of this study, potential sources of overcorrection could be the coincidental existence of class ‘S’ (and class ‘CR’) and sun glint event in the shallow areas as these substrates often reach near or above water surface.’

L588: “exhibits … leads”.

RE: The revised sentence now reads as ‘Strategy-1 exhibits nearly flat spectra profiles, resulting in reduced performance in the classification of benthic habitats.’

L599-600: “performs better … and results in reliable”.

RE: The revised sentence reads as ‘Comparison of the four glint correction algorithms when applied to high-resolution UAV imagery following whole image (Strategy-1) and glint-impacted (Strategy-2) strategies shows that the image pre-processed with Strategy-2 and Lyzenga et al. (2006) performs better than the other methods and results in reliable maps of corals.’

Round 2

Reviewer 1 Report

The authors have gone some way to address my previous queries but the manuscript still has some issues.

First, the authors don't seem to fully understand the difference (or lack of it) between the Lyzenga, Joyce and Hedley methods, even though this is very clearly spelled out in Kay et al. 2009, and their own results show it. In table 5 many of the results are numerically identical between these methods. The conclusion should be that they perform similarly, looks like just a few % difference on percentages that are 80-100%. Therefore the statement in the abstract about the Lyzenga method doesn't stand up, the conclusion should be just about strategy 2. Under the scientific method one is supposed to test a hypothesis according to a statistical test. Even if that isn't done the basic principle should be remembered, that +few% difference on numbers varying from 80-100% probably doesn't mean anything. Also the revised explanation of these methods is very poor (below I suggest how it could be expressed).

Strategy 2 still isn't described, references to other papers are given. Does that mean that Strategy is already published? For me, it is the main aspect of interest in this paper. There is still an issue with the discontinuity it introduces. The plots in Fig. 4 are useless they just show a massive discontinuity which I consider an artefact of the method - for sure it is not "real". These plots show reflectances approaching 1, there cannot be reflectances of this magnitude, it's just nonsense. I expect this arises because of the normalisation, why normalise? What purpose does it serve? Most classification schemes can handle normalisation themselves, or if necessary it could be done before the classification, not presented in a plot as "reflectances". Glint correction should just remove the glint, not introduce large discontinuities. At the border between where the glint is applied and where it isn't is presumably an area of low glint, so the plot from no-glint correction to glint correction should be seamless, or with a small discontinuity, maybe. Such a plot would actually be really helpful to see how the glint correction is working.

These are my recommendations to improve the paper, I think they are easily addressed so I would class them as as "minor", but I'd also rather not review the paper a third time.

Some other specific comments:

"RE: Yes it has pointed out in the Introduction section. The added sentence is ‘There are few studies of sun glint correction methods applied to UAV camera imagery across coral reef areas.’"

Fine - but could the authors point out some things that are different about UAV imagery compare to other types of imagery to which these methods are normally applied? If none there'd be no need to do these tests. Perhaps the roll and yaw of the drone, the fact that images are stitched together (I assume), are relevant? It would be good if the authors could think about this a bit and add something.

Table 1 - I don't believe reference 27 used IKONOS data. Also the notes include abbreviations not included and exclude some that are.

Line 107 still says the methods are predominately applied to satellite images, yet in the correct version of Table 1 only 4 of 10 entries are satellite data.

Line 130 - after the word "manage" please put a reference to the papers for methods of deglinting of high spatial resolution imagery which require "wind speed and direction, wave slope and other parameters at the time of satellite overpass". If there are none delete the sentence.

"RE: This is due to transition between glint-corrected and uncorrected images when implemented under Strategy-2 and it also explained inside the manuscript. Previous studies including Kay et al. 2009 did not apply Strategy-2 and therefore, results are not comparable."

It is not about "Strategy 2" it is about the comparison between where glint correction is applied and where it isn't. When glint correction is applied the reflectances should be a bit less than when it isn't because you have removed the glint component of the reflectance. The reflectance should not increase massively. Reflectances close to 1 are non-physical. Kay et al. 2009 shows many plots where these same glint correction methods have been applied. The reflectances are not close to 1. The problem is in the implementation of your glint correction methods, not Strategy 2. I think you need to remove the normalisation, it makes everything very hard to interpret.

Line 286. The explanation of the Lyzenga method is woeful, eq 2 is just the expression for the covariance, it isn't explained how it used. Kay et al 2009 says this:
"Note that because Hedley et al. use a least squares method to calculate the regression slope, rij (in Equation 37) and bi (in Equation 34) are the same. The two methods are equivalent, except that Hedley et al. use the minimum value of the NIR radiance in the final term, where Lyzenga et al. use the mean value."
Assuming this is correct, there's no need to show the covariance equation, its just a more complicated way of calculating the regression (most of the coefficients are not used). The only difference is in the subtracted NIR value.

Line 292 I would say the the approach is "the same" not "no exception"

Line 294 The mode is the most common value, not the maximum as stated here. With digital numbers you could literally find the mode, but if the data is considered continuous (not integers) it's a bit more tricky, one way is to have the concept of a distribution of which the mode is the value at the highest peak, or it could be done by some kind of numerical histogram method. The authors need to fix this as at the moment its unclear and incorrect. Is it even the mode that is used?

Overall this part has been restructured very badly, it could very much clearer. Why not just give this equation:

R(i)' = R(i) - b(i)[R(NIR) - R_ref(NIR)]

and explain that for the three methods R_ref is either the min, mean or mode (or max, whatever it actually was). And that value was derived over THE SAME sample pixel area in each case. I hope it was the same set of pixels, because even for one method performance very much depends on the set of sample pixels chosen. However the results show quite well that these three methods give similar results, which means this is less of an issue.

Author Response

The authors have gone some way to address my previous queries but the manuscript still has some issues.

First, the authors don't seem to fully understand the difference (or lack of it) between the Lyzenga, Joyce and Hedley methods, even though this is very clearly spelled out in Kay et al. 2009, and their own results show it. In table 5 many of the results are numerically identical between these methods. The conclusion should be that they perform similarly, looks like just a few % difference on percentages that are 80-100%. Therefore the statement in the abstract about the Lyzenga method doesn't stand up, the conclusion should be just about strategy 2. Under the scientific method one is supposed to test a hypothesis according to a statistical test. Even if that isn't done the basic principle should be remembered, that +few% difference on numbers varying from 80-100% probably doesn't mean anything. Also the revised explanation of these methods is very poor (below I suggest how it could be expressed).

Strategy 2 still isn't described, references to other papers are given. Does that mean that Strategy is already published? For me, it is the main aspect of interest in this paper. There is still an issue with the discontinuity it introduces. The plots in Fig. 4 are useless they just show a massive discontinuity which I consider an artefact of the method - for sure it is not "real". These plots show reflectances approaching 1, there cannot be reflectances of this magnitude, it's just nonsense. I expect this arises because of the normalisation, why normalise? What purpose does it serve? Most classification schemes can handle normalisation themselves, or if necessary it could be done before the classification, not presented in a plot as "reflectances". Glint correction should just remove the glint, not introduce large discontinuities. At the border between where the glint is applied and where it isn't is presumably an area of low glint, so the plot from no-glint correction to glint correction should be seamless, or with a small discontinuity, maybe. Such a plot would actually be really helpful to see how the glint correction is working.

RE: We suggested Lyzenga et al. [48] following Strategy-2, NOT Lyzenga et al. [48] alone.

Some texts revised for clarity: “All the glint corrected images are normalized to provide a simulated positive 0 to 1 water-leaving reflectance value. The reflectance values in Figures 3 and 4 are not the "true" benthic reflectance; normalization has been performed to observe the effects of glint removal on reflectance.”

These are my recommendations to improve the paper, I think they are easily addressed so I would class them as as "minor", but I'd also rather not review the paper a third time.

Some other specific comments:

"RE: Yes it has pointed out in the Introduction section. The added sentence is ‘There are few studies of sun glint correction methods applied to UAV camera imagery across coral reef areas.’"

Fine - but could the authors point out some things that are different about UAV imagery compare to other types of imagery to which these methods are normally applied? If none there'd be no need to do these tests. Perhaps the roll and yaw of the drone, the fact that images are stitched together (I assume), are relevant? It would be good if the authors could think about this a bit and add something.

RE: A sentence added to the text: “The internal (focal length, principal point, lens distortion) and external orientations (X-Y-Z spatial positions, omega-phi-kappa angular positions) indicating stability of the drone are all taken into account before UAV imagery is stitched together by a robust photogrammetric approach.”

Table 1 - I don't believe reference 27 used IKONOS data. Also the notes include abbreviations not included and exclude some that are.

RE: Corrected; it should be CASI.

Line 107 still says the methods are predominately applied to satellite images, yet in the correct version of Table 1 only 4 of 10 entries are satellite data.

RE: added the word “airborne”.

Line 130 - after the word "manage" please put a reference to the papers for methods of deglinting of high spatial resolution imagery which require "wind speed and direction, wave slope and other parameters at the time of satellite overpass". If there are none delete the sentence.

RE: Added two references [36,44].

"RE: This is due to transition between glint-corrected and uncorrected images when implemented under Strategy-2 and it also explained inside the manuscript. Previous studies including Kay et al. 2009 did not apply Strategy-2 and therefore, results are not comparable."

It is not about "Strategy 2" it is about the comparison between where glint correction is applied and where it isn't. When glint correction is applied the reflectances should be a bit less than when it isn't because you have removed the glint component of the reflectance. The reflectance should not increase massively. Reflectances close to 1 are non-physical. Kay et al. 2009 shows many plots where these same glint correction methods have been applied. The reflectances are not close to 1. The problem is in the implementation of your glint correction methods, not Strategy 2. I think you need to remove the normalisation, it makes everything very hard to interpret.

RE: We reiterate Kay et al. 2009 didn’t apply Strategy-2. We presented results we obtained and explained the probable reasons.

Line 286. The explanation of the Lyzenga method is woeful, eq 2 is just the expression for the covariance, it isn't explained how it used. Kay et al 2009 says this:

"Note that because Hedley et al. use a least squares method to calculate the regression slope, rij (in Equation 37) and bi (in Equation 34) are the same. The two methods are equivalent, except that Hedley et al. use the minimum value of the NIR radiance in the final term, where Lyzenga et al. use the mean value."

Assuming this is correct, there's no need to show the covariance equation, its just a more complicated way of calculating the regression (most of the coefficients are not used). The only difference is in the subtracted NIR value.

Line 292 I would say the the approach is "the same" not "no exception"

RE: revised the whole subsection.

Line 294 The mode is the most common value, not the maximum as stated here. With digital numbers you could literally find the mode, but if the data is considered continuous (not integers) it's a bit more tricky, one way is to have the concept of a distribution of which the mode is the value at the highest peak, or it could be done by some kind of numerical histogram method. The authors need to fix this as at the moment its unclear and incorrect. Is it even the mode that is used?

Overall this part has been restructured very badly, it could very much clearer. Why not just give this equation:

R(i)' = R(i) - b(i)[R(NIR) - R_ref(NIR)]

and explain that for the three methods R_ref is either the min, mean or mode (or max, whatever it actually was). And that value was derived over THE SAME sample pixel area in each case. I hope it was the same set of pixels, because even for one method performance very much depends on the set of sample pixels chosen. However the results show quite well that these three methods give similar results, which means this is less of an issue.

RE: Revised texts now read as: First, we applied the three methods: proposed by Hedley et al. [28], Lyzenga et al. [48] and Joyce [49]. All these methods used a subset of deep-water pixel samples (see Figure 3 for yellow ROI) to compute the regression slope () from a line of linear correlation between NIR (R(NIR)) and each visible band ((VIS)) (as expressed in Equation 1). Note that Hedley et al. [28] and Joyce [49] used the least square approach to define the  whereas Lyzenga et al. [48] used the covariance between each visible band and the NIR for obtaining .

                                          (1)

where  is the minimum in NIR value used for Hedley et al. [28], the mean NIR value for Lyzenga et al. [48] and the modal NIR value for Joyce [49]. These values were derived from the same set of deep-water pixels.”

Reviewer 3 Report

The manuscript has been improved from the first version. However, some problems still remain.

The authors did not reduce significantly the Introduction, with Table 1 a bit shortened, but still there. To improve readability, such table should be moved to an appendix or within the supplementary materials.

As already suggested in the first-round review, the comparison between the presented results and those related to other sources of video imagery (e.g. satellite images), already presented in the recent literature, could be worth of investigation and should be included in the Discussion section.

Author Response

The manuscript has been improved from the first version. However, some problems still remain.

The authors did not reduce significantly the Introduction, with Table 1 a bit shortened, but still there. To improve readability, such table should be moved to an appendix or within the supplementary materials.

RE: Table 1 moved to Supplementary Materials and introduction has been shortened.

As already suggested in the first-round review, the comparison between the presented results and those related to other sources of video imagery (e.g. satellite images), already presented in the recent literature, could be worth of investigation and should be included in the Discussion section.

RE: We are not clear what the reviewer indicating to discuss. Note, underwater video imagery used for coral mapping should not experience sun glint affects.